# Spatiotemporal organisation of human sensorimotor beta burst activity

Catharina Zich[1,2,3]*, Andrew J Quinn[4,5], James J Bonaiuto[6,7], George O'Neill[8], Lydia C Mardell[1], Nick S Ward[1], Sven Bestmann[1,8]

[1]Department of Clinical and Movement Neuroscience, UCL Queen Square Institute of Neurology, London, United Kingdom; [2]Wellcome Centre for Integrative Neuroimaging, FMRIB, Nuffield Department of Clinical Neurosciences, University of Oxford, Oxford, United Kingdom; [3]Medical Research Council Brain Network Dynamics Unit, University of Oxford, Oxford, United Kingdom; [4]Oxford Centre for Human Brain Activity, Wellcome Centre for Integrative Neuroimaging, Department of Psychiatry, University of Oxford, Oxford, United Kingdom; [5]Centre for Human Brain Health, School of Psychology, University of Birmingham, Birmingham, United Kingdom; [6]Institut des Sciences Cognitives Marc Jeannerod, CNRS UMR 5229, Bron, France; [7]Université Claude Bernard Lyon 1, Université de Lyon, Lyon, France; [8]Wellcome Centre for Human Neuroimaging, Department of Imaging Neuroscience, UCL Queen Square Institute of Neurology, London, United Kingdom

*For correspondence:
catharina.zich@ndcn.ox.ac.uk

**Competing interest:** The authors declare that no competing interests exist.

**Abstract** Beta oscillations in human sensorimotor cortex are hallmark signatures of healthy and pathological movement. In single trials, beta oscillations include bursts of intermittent, transient periods of high-power activity. These burst events have been linked to a range of sensory and motor processes, but their precise spatial, spectral, and temporal structure remains unclear. Specifically, a role for beta burst activity in information coding and communication suggests spatiotemporal patterns, or travelling wave activity, along specific anatomical gradients. We here show in human magnetoencephalography recordings that burst activity in sensorimotor cortex occurs in planar spatiotemporal wave-like patterns that dominate along two axes either parallel or perpendicular to the central sulcus. Moreover, we find that the two propagation directions are characterised by distinct anatomical and physiological features. Finally, our results suggest that sensorimotor beta bursts occurring before and after a movement can be distinguished by their anatomical, spectral, and spatiotemporal characteristics, indicating distinct functional roles.

## Editor's evaluation

This paper provides important insights into the spatial organization of β-oscillatory activity in the human brain, which is a crucial dynamic feature of frontal and parietal networks involved in movement preparation and sensory prediction. Using high-resolution source reconstruction with Magnetoencephalography in humans, the authors provide compelling evidence demonstrating that β oscillations are organized as travelling waves in two distinct directions relative to the central sulcus. Furthermore, the study convincingly shows that the spatiotemporal organization of β bursts is systematically linked to behavior, specifically motor execution. These findings have important implications for our understanding of the neural mechanisms that underlie movement planning and execution in the human brain.

## Introduction

Neural activity at the rate of 13–30 Hz constitutes one of the most prominent electrophysiological signatures in the sensorimotor system (*Baker, 2007*; *Brown, 2007*). This sensorimotor beta activity is traditionally seen to reflect oscillations: sustained rhythmic synchronous spiking activity within neural populations. However, a substantial proportion of sensorimotor beta activity occurs in bursts of intermittent, transient periods of synchronous spiking activity (*Jones, 2016*) which relate to both motor, perceptual, and sensory function (*Enz et al., 2021*; *Feingold et al., 2015*; *Heideman et al., 2020*; *Sherman et al., 2016*; *Shin et al., 2017*; *Sporn et al., 2020*; *Tinkhauser et al., 2017a*; *Wessel, 2020*; *Zich et al., 2018*) and pathophysiological movement (*Cagnan et al., 2019*; *Deffains et al., 2018*; *Tinkhauser et al., 2017a*; *Tinkhauser et al., 2017b*), but their functional role remains unclear.

Sensorimotor beta burst activity is commonly considered as zero-lagged (or standing wave) activity which is generated by the summation of synchronised layer-specific inputs within cortical columns that result in a cumulative dipole with a stereotypical wavelet shape in the time domain (*Bonaiuto et al., 2021*; *Law et al., 2022*; *Neymotin et al., 2020*). These time periods of synchronous activity which generate standing wave activity are thought to convey little information encoding (*Brittain and Brown, 2014*; *Carhart-Harris, 2018*; *Carhart-Harris et al., 2014*). This view sides with the proposed akinetic role of high sensorimotor beta states (*Gilbertson et al., 2005*; *Joundi et al., 2012*; *Khanna and Carmena, 2017*; *Pogosyan et al., 2009*). However, burst activity may have heterogeneous and mechanistically distinct components which can be characterised by their distinct spatial, temporal, and spectral structure (*Law et al., 2022*; *Zich et al., 2020*) that, in addition to zero-lagged activity, contains spatiotemporal gradients, or travelling wave, components.

In animals, for example, a high proportion of sensorimotor beta activity occurs as travelling waves (*Rubino et al., 2006*; *Rule et al., 2018*), in addition to highly synchronous standing waves. In travelling waves, the relative timing of fluctuations of synchronous spiking activity is not precisely zero-lagged but adopts a phase offset and moves across space. Propagation of neural activity constitutes one mechanism for cortical information transfer and travelling waves have been described over spatial scales that range from the mesoscopic (single cortical areas and millimetres of cortex) to the macroscopic (global patterns of activity over several centimetres) and extend over temporal scales from tens to hundreds of milliseconds (*Alexander et al., 2019*; *Davis et al., 2021*; *Heitmann et al., 2017*; *Muller et al., 2018*; *Roberts et al., 2019*; *Rule et al., 2018*).

Characterising travelling wave components within sensorimotor beta burst activity is of relevance as it would provide insights into the putative underlying mechanisms and functional roles of sensorimotor beta activity. For instance, in general terms, spatiotemporal propagation of high-amplitude beta may support information transfer across space and may reflect the spatiotemporal patterns of sequential activation required for movement initiation (*Best et al., 2017*; *Rubino et al., 2006*). At the macro-scale level, the specific propagation properties, such as propagation direction and speed, may provide further constraints for the putative functional role of burst activity in organising behaviour across different brain regions (*Ding and Ermentrout, 2021*), including the modulation of neural sensitivity (*Davis et al., 2020*) or the sequencing of muscle representations in motor cortex (*Muller et al., 2018*; *Riehle et al., 2013*; *Takahashi et al., 2015*). In humans, the precise properties of beta bursts and whether their high-amplitude activity comprise distinct spatiotemporal gradients remain unclear.

To address this, we here employed high signal-to-noise (SNR) magnetoencephalography (MEG) in healthy human subjects during simple visually cued motor behaviour. We show that beta burst activity in sensorimotor cortex occurs in planar spatiotemporal wave-like patterns that dominate along two anatomical axes. Crucially, our results show structure beyond the inherent limitations of source reconstruction such as volume conduction or the spatial pattern of beamformer weights. Moreover, we find that the two propagation directions are characterised by distinct anatomical and physiological features. Finally, to further extend our understanding of the functional roles of sensorimotor beta bursts we compare bursts occurring before and after a movement. Our results suggest that sensorimotor beta bursts occurring before and after a movement can be distinguished by their anatomical, spectral, and spatiotemporal characteristics, indicating distinct functional roles.

## Results

### Temporal, spectral, and spatial burst characteristics

Participants completed three blocks per recording session, and 1–5 sessions on different days. We analysed 123–611 trials per participant ($M$ = 438.5, SD = 151.0 across individuals) in which correct key presses were made with either the right index or middle finger, in response to congruent imperative stimuli and high coherence visual cues (see 'Participants and experimental task'; *Bonaiuto et al., 2018*; *Little et al., 2019*). We focussed on these trial types to delineate the multi-dimensional (temporal, spectral, spatial) properties of sensorimotor beta burst activity (*Figure 1a and b*; *Zich et al., 2020*). Bursts were identified over a 4 s time window (−2–2 s relative to the button press), in the beta frequency range (13–30 Hz) and a region of interest (ROI) spanning the primary motor cortex (M1) and adjacent areas of the primary sensory cortex and premotor cortex using session-specific amplitude thresholding (*Little et al., 2019*) and 5D clustering (see 'Burst operationalisation', Figure 6).

First, we characterise several first-level burst characteristics (see 'Burst characteristics'). In the temporal domain, we observed the expected increase in burst probability pre- vs. post-movement (*Figure 1c*). Burst duration was consistent across subjects ($M$ = 238 ms, SD = 23 ms across individuals, temporal resolution 50 ms, *Figure 1e*). Spectrally, while beta bursts occurred throughout the beta frequency range, most bursts were identified in the lower beta frequency range (*Figure 1c*), with a consistent frequency spread across subjects ($M$ = 3 Hz, SD = 0 Hz across individuals, frequency resolution 1 Hz, *Figure 1e*). To examine burst probability as a function of space across subjects, individual subject maps were spatially normalised, projected onto a single surface, and then averaged across subjects. Topographically, bursts were most likely to occur in M1 (*Figure 1d*, see *Figure 1—figure supplement 1* for individual subject maps) and spanned, on average, 10% of the ROI's surface area (apparent spatial width: $M$ = 6 cm²; SD = 0.9 cm² across individuals). Post hoc analysis revealed no significant differences between beta bursts around index or middle finger movement.

We performed a range of control analyses to examine whether our results can be explained by trivial properties of the beamformer itself. Firstly, we sought to assess whether differences in the bursts' apparent spatial width could be explained by differences in SNR across and/or within sessions rather than differences in the spatial distribution of cortical activity. We reasoned that if differences in SNR across sessions would explain bursts' apparent spatial width, then burst amplitude and burst apparent spatial width should be negatively correlated (for a schematic illustration, see *Figure 1—figure supplement 2ai*). The absence of significant correlations between burst amplitude and burst apparent spatial width, both across sessions within subjects and also across sessions and individuals (*Figure 1—figure supplement 2aii*), suggests that the apparent spatial width of bursts is not solely explained by the differences in SNR across sessions and across individuals.

Further we reasoned that if the apparent spatial width is driven by differences in SNR across bursts within a session, then a positive relationship between burst amplitude and burst apparent spatial width within sessions should be present, and there should be no systematic phase differences across different spatial locations within each burst (for a schematic illustration, see *Figure 1—figure supplement 2b*). While burst amplitude and burst apparent spatial width are positively correlated within sessions (Pearson's $r$: $M$ = 0.749, SD = 0.056 across sessions, all ps<0.001), we consistently observed diverse phase lags across space within these bursts (see 'Sensorimotor burst activity propagates along one of two axes'), which are unlikely to arise simply from amplitude scaling of a single source.

Together, these control analyses suggest that differences in bursts' apparent spatial width is not merely due to differences in SNR across and/or within sessions, but for the most part due to differences in the spatial distribution of cortical activity.

### Sensorimotor beta burst activity is propagating

The precise decomposition of beta bursts into their spectral, spatial, and temporal signal domains allowed us to next assess any spatiotemporal gradients within sensorimotor beta bursts. For each burst, we identified the dominant propagation direction and propagation speed. Propagation direction and speed were estimated from critical points in the oscillatory cycle (*Figure 2a*) and then averaged across critical points within one burst. The propagation direction at each critical point was estimated from the relative latency (*Figure 2bi*). Next, using linear regression (*Balasubramanian et al., 2020*), whereby the relative latency at the surface location was predicted from the coordinates of the surface

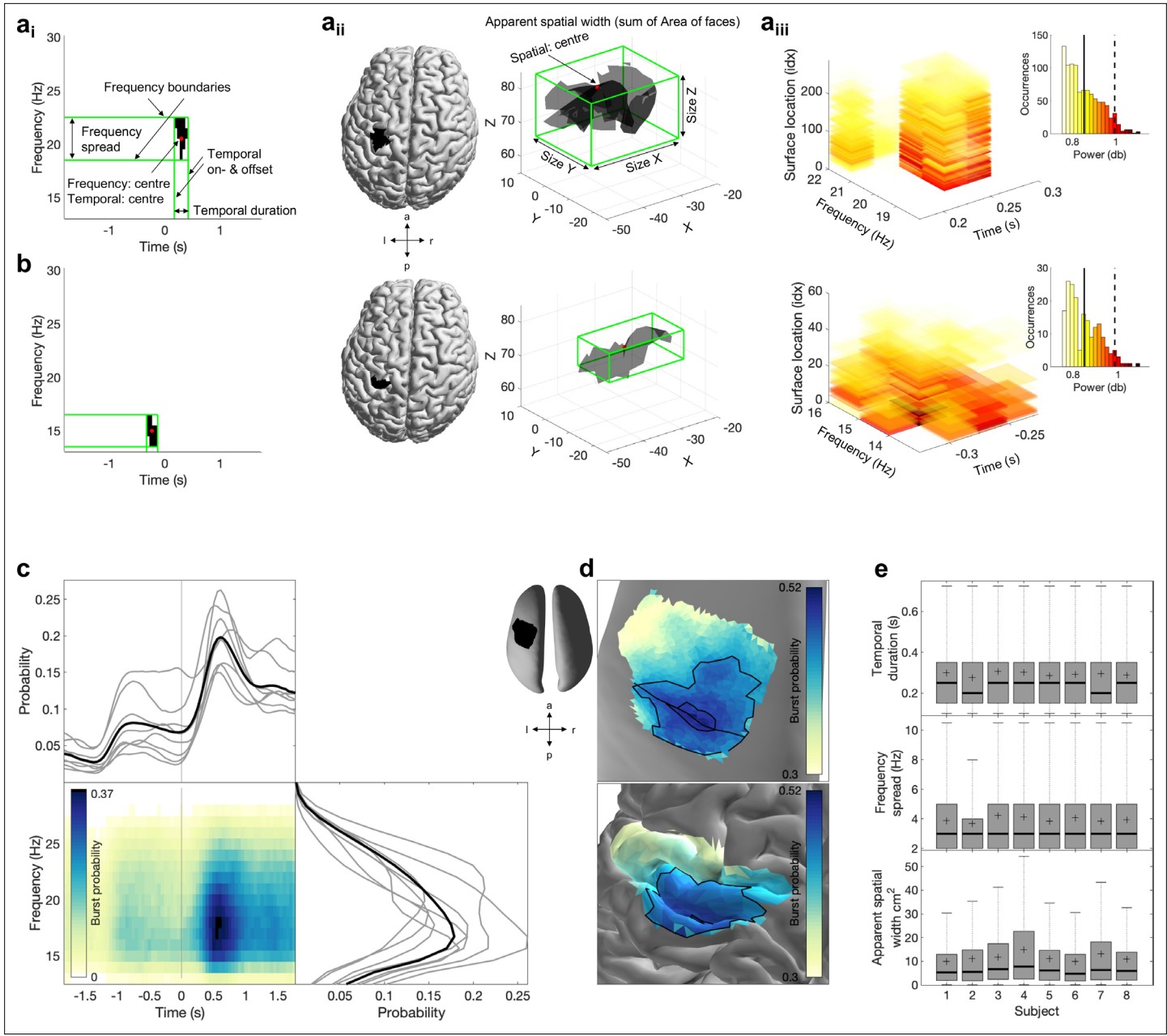

**Figure 1.** Spectral, temporal, and spatial beta burst characteristics. (**a**) Burst characteristics for a single example burst. (**a$_i$**) Temporal and spectral burst characteristics. (**a$_{ii}$**) Spatial burst characteristics. (**a$_{iii}$**) Burst amplitude. Shown is the amplitude for each temporal, spectral, and spatial location of that burst, as well as the histogram across all three signal domains. Note that we use integer indexation of 3D surface location (idx) for visualisation here. All other analyses use the actual 3D surface location as provided by Cartesian x,y,z coordinates. Usually the mean (straight line) or the 95 percentile (dashed line) is reported as burst amplitude. (**b**) Same as (**a**) for a different burst. (**c**) Burst probability (i.e. number of bursts relative to the number of epochs) as a function of time and frequency across all bursts of all subjects (see *Figure 1—figure supplement 1* for individual subjects). Burst probability as a function of time (bottom) and frequency (left) is shown for each subject separately (grey lines) and across subjects (black line). (**d**) Burst probability as a function of space across all bursts of all subjects on the inflated surface (top) and original surface (bottom). Highlighted are the central sulcus and the borders for 0.44 and 0.49 burst probability. To visualise burst probability as a function of space across subjects, individual subject maps were spatially normalised, projected onto a single surface, and then averaged across subjects. *Figure 1—figure supplement 1* depicts the burst probability for each subject in native space. (**e**) Burst temporal duration (top), frequency spread (middle), and apparent spatial width (bottom) for each subject as boxplot.

The online version of this article includes the following figure supplement(s) for figure 1:

**Figure supplement 1.** Beta power and burst probability are shown for all three signal domains for each subject.

**Figure supplement 2.** Schematic illustration of LCMV-related aspects for 3D bursts.

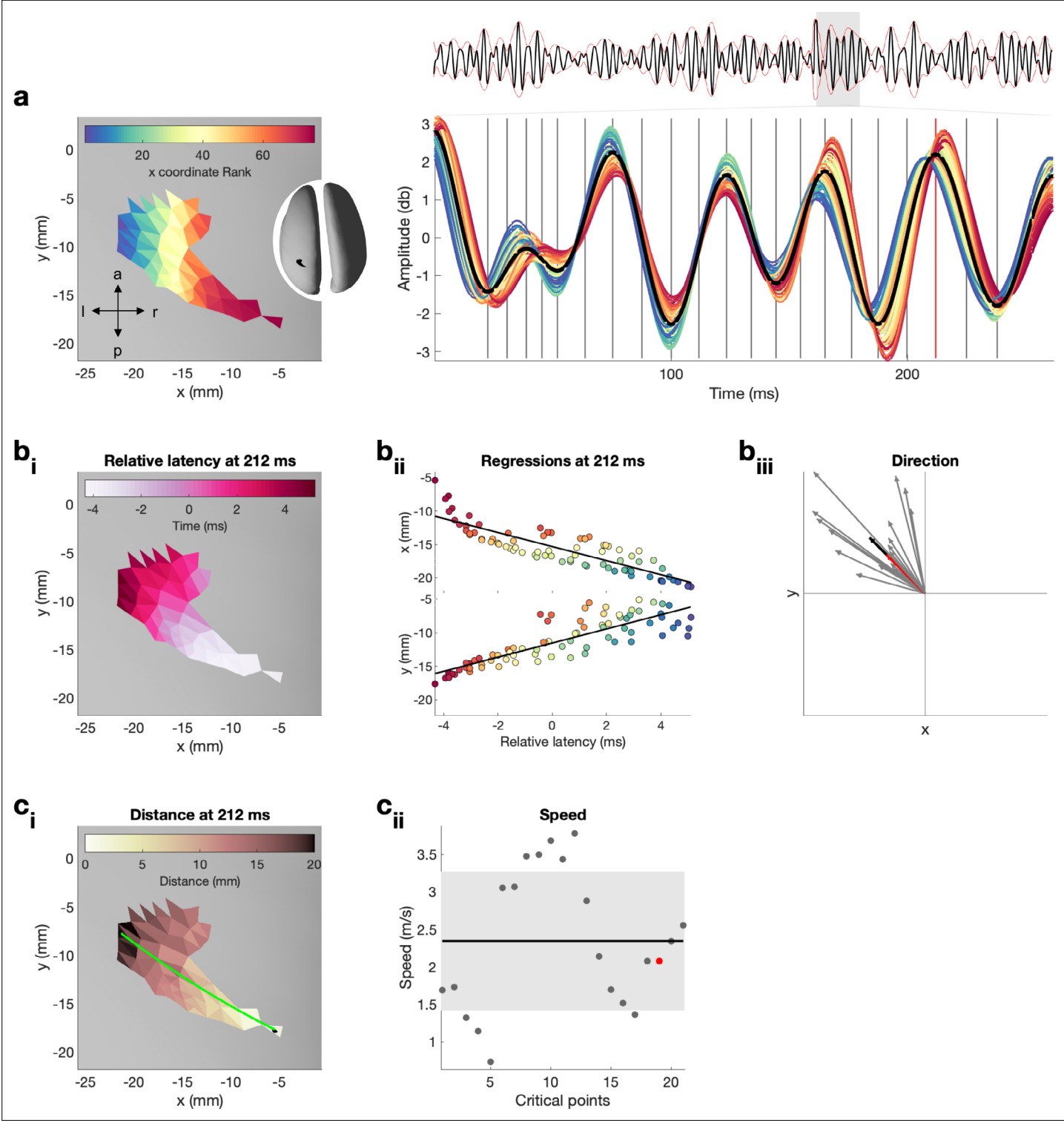

**Figure 2.** Quantification of propagation direction and propagation speed on one exemplar burst. For a dynamic version, that is, updated for each critical point, see *Figure 2—video 1*. (**a**) Left: single burst on inflated surface. Spatial locations are colour-coded by their x coordinate rank. Note that x coordinate rank is only used here to illustrate the spatial location of the neural time series shown on the right. Right: neural activity in the beta range (13–30 Hz) from each surface location for the temporal duration of the entire epoch (top, average across spatial locations in black and amplitude-envelope in red) and in large for the temporal duration of the burst. Vertical lines indicate critical points (four critical points per oscillatory cycle, i.e. peak and trough as well as peak-trough and trough-peak midpoint) at which propagation direction and propagation speed were estimated. Red vertical line indicates the control point at 212 ms, shown in (**b**$_{i,ii}$) and (**c**$_{i,ii}$), and highlighted in (**b**$_{iii}$) and (**c**$_{iii}$). (**b**$_i$) Relative latencies of the critical point at 212 ms

*Figure 2 continued on next page*

*Figure 2 continued*

as a function of space illustrated on inflated surface. (**b**ᵢᵢ) Simple linear regressions between latency at surface location and x (top) as well as y (bottom) coordinates of the surface location for the critical point at 212 ms. Colour refers to the x coordinate rank as illustrated in (**a**). (**b**ᵢᵢᵢ) Propagation direction obtained from regression coefficients for each critical point (grey), the critical point at 212 ms (red), and the average across all critical points (black, i.e. propagation direction of burst). (**c**ᵢ) Distance, that is, exact geodesic distance, from the surface location with the smallest relative latency to each surface location on the inflated surface for the critical point at 212 ms. Green line indicated the path, that is, distance, from the surface location with the smallest to the surface location with the largest relative latency. (**c**ᵢᵢ) Propagation speed for each critical point (grey), the critical point at 212 ms (red). The standard deviation across critical points is indicated by the grey area, and the average across all critical points (i.e. propagation speed of burst) is indicated by the black horizontal line.

The online version of this article includes the following video and figure supplement(s) for figure 2:

**Figure supplement 1.** Examples of bursts with (**a**) one propagation direction and (**b, c**) complex propagation patterns.

**Figure supplement 2.** Propagation speed using the distance on the original or the inflated surface.

**Figure 2—video 1.** Same as *Figure 2*, but each frame corresponds to a different critical point within the burst.

https://elifesciences.org/articles/80160/figures#fig2video1

location of the inflated surface (see 'Propagation direction and speed of neural activity within bursts'). We excluded complex spatiotemporal patterns such as random or circular patterns (*Figure 2—figure supplement 1b and c*; *M* = 8.25%, SD = 0.88% across individuals; *Denker et al., 2018*; *Rule et al., 2018*), to focus on bursts with a dominant planar propagation orientation (*Figure 2—figure supplement 1a*; *M* = 79.6%, SD = 2.4% across individuals; *Balasubramanian et al., 2020*; *Rubino et al., 2006*; *Takahashi et al., 2011*).

To test whether the planar spatiotemporal structure of bursts is significant, we compared the propagation properties detected in real burst activity to those of surrogate data for a subset of 100 randomly selected bursts sampled across all subjects. For each burst, 1000 phase-randomised surrogates (*Hurtado et al., 2004*) were created and the propagation properties of the real data were compared to their distribution from 1000 surrogates (see **'**Statistical analysis'). Real sensorimotor beta burst activity exhibited significantly stronger planar spatiotemporal structure than spectrally matched surrogate data (all 100 bursts p<0.01).

## Accuracy of the propagation direction estimation in simulated and surface meshes

Before assessing the propagation direction of sensorimotor beta burst activity, we evaluated the accuracy of the propagation direction estimation (see 'Accuracy of the propagation direction detection in simulated and real meshes'). To this end, we created 360 noise-free high-resolution gradients spanning 1–360° (in steps of 1°; subset shown in *Figure 3a*). To tease apart inaccuracies due to the method from inaccuracies due to the nature of the mesh, we first estimated the propagation direction of these gradients from three different 2D surface mesh types (square mesh, circular mesh, random mesh; *Figure 3b–d*). Further, we evaluate the accuracy using three spatial sampling rates (N/2, N, N × 2, whereby N approximates the spatial sampling of the real mesh), as the real mesh is irregular with varying spatial sampling across the mesh. By comparing the true and estimated propagation direction, we found that under noise-free conditions, the propagation direction can be estimated accurately from regular meshes (*Figure 3b and c*), whereas the mean error is roughly twice as large for random meshes (*Figure 3d*). Further, for the random mesh a linear relationship between accuracy and spatial sampling can be observed.

This is relevant because the surface mesh obtained from brain imaging data is irregular. When evaluating the accuracy of the propagation direction estimates using the real mesh (*Figure 3e*) and the real spatial burst properties, we found an average error of 7° between true and estimated propagation direction, with little variability across individuals (SD = 0.5° across subjects) and angles (SD = 1.5° across angles; *Figure 3ei*). Across individuals the error was smallest for gradient directions around 104/284° and largest around 170/350° (*Figure 3ei*). Further, for bursts with a larger apparent spatial width (i.e. containing more spatial samples), the estimated error is lower (*Figure 3eii*).

Overall, these results suggest that propagation directions can be estimated with sufficient accuracy from higher SNR MEG recordings over a relatively small cortical patch, as here.

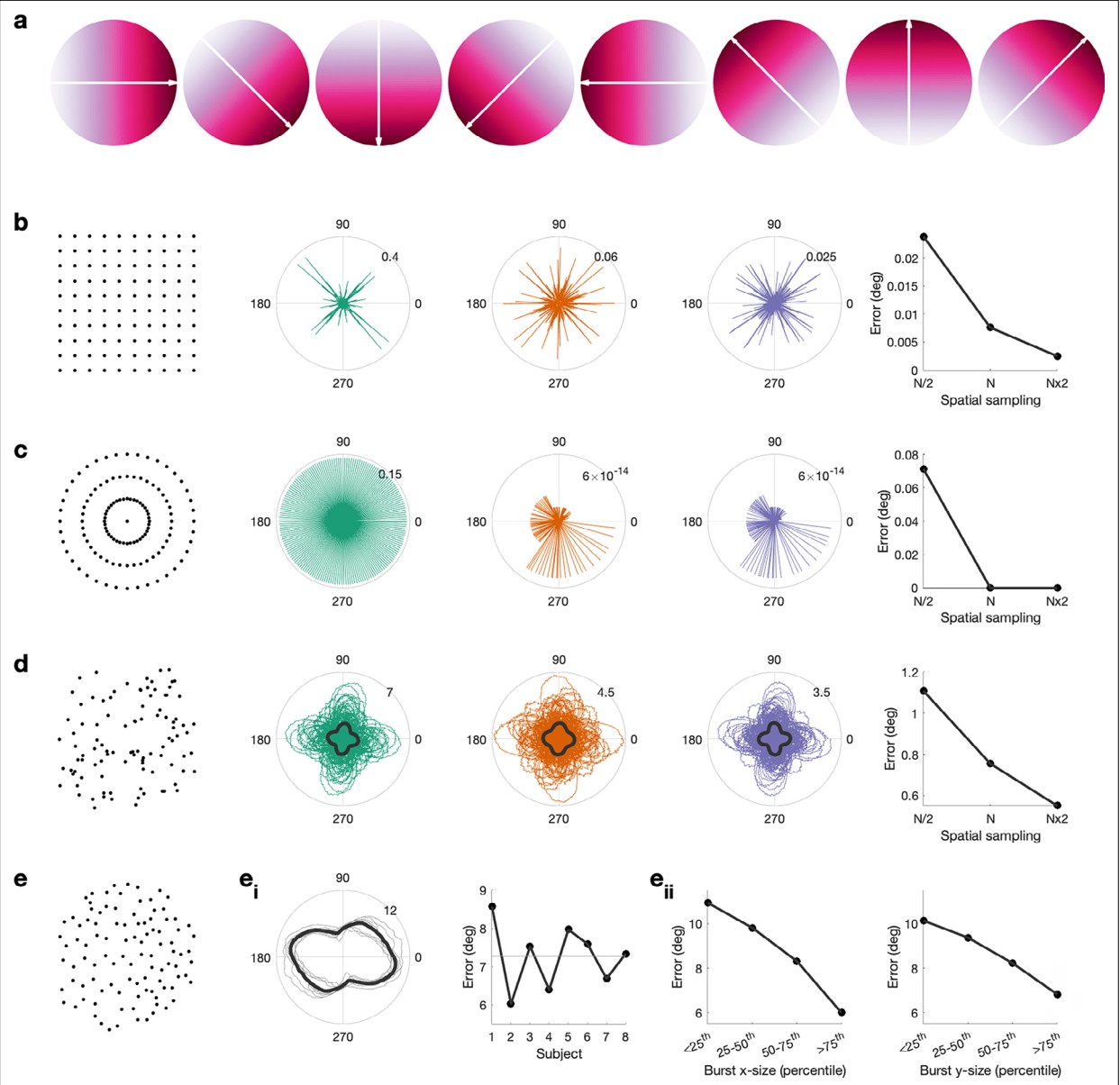

**Figure 3.** Propagation direction can be estimated accurately from cortical meshes. (**a**) Simulated gradient at 0, 45, 90, 135, 180, 225, and 270°. (**b**) Error between simulated gradient and estimated gradient on a square mesh. We test three different spatial sampling rates, N/2 (green), N (red), and N × 2 (blue), whereby the spatial sampling of N is roughly equivalent to the spatial sampling of the inflated surface. Error is calculated for 1–360° in steps of 1°. The median error per spatial sampling is shown in the right, that is, higher spatial sampling results in a lower error. (**c**) As (**b**), but for a circular mesh. (**d**) Error between simulated gradient and estimated gradient on a random mesh. 100 random meshes were generated. The error for each iteration is shown as well as the mean across iterations (black line). (**e**) Error between simulated gradient and estimated gradient on the inflated surface. The error was calculated for each burst. (**e**$_i$) The mean across bursts is shown for each subject (grey lines) and across subjects (black line). For each subject, the mean error across all angles and bursts is shown, that is, error is comparable across subjects. (**e**$_{ii}$) Error as a function of burst size along the x-axis (left) and y-axis (right), that is, the error is lower in bigger burst.

## Sensorimotor burst activity propagates along one of two axes

Having established that the spatial sampling of the cortical mesh is sufficient to detect propagation in simulated gradients, we analysed the propagation properties of the sensorimotor beta burst activity. We observed that neural activity within beta bursts propagates along one of two dominant axes, which were approximately 90° apart (*Figure 4a*): one anterior–posterior (a-p) axis traversing the central sulcus in approximately perpendicular fashion, and one medial–lateral (m-l) axis running approximately parallel to (i.e. along) the central sulcus. The propagation distributions along these

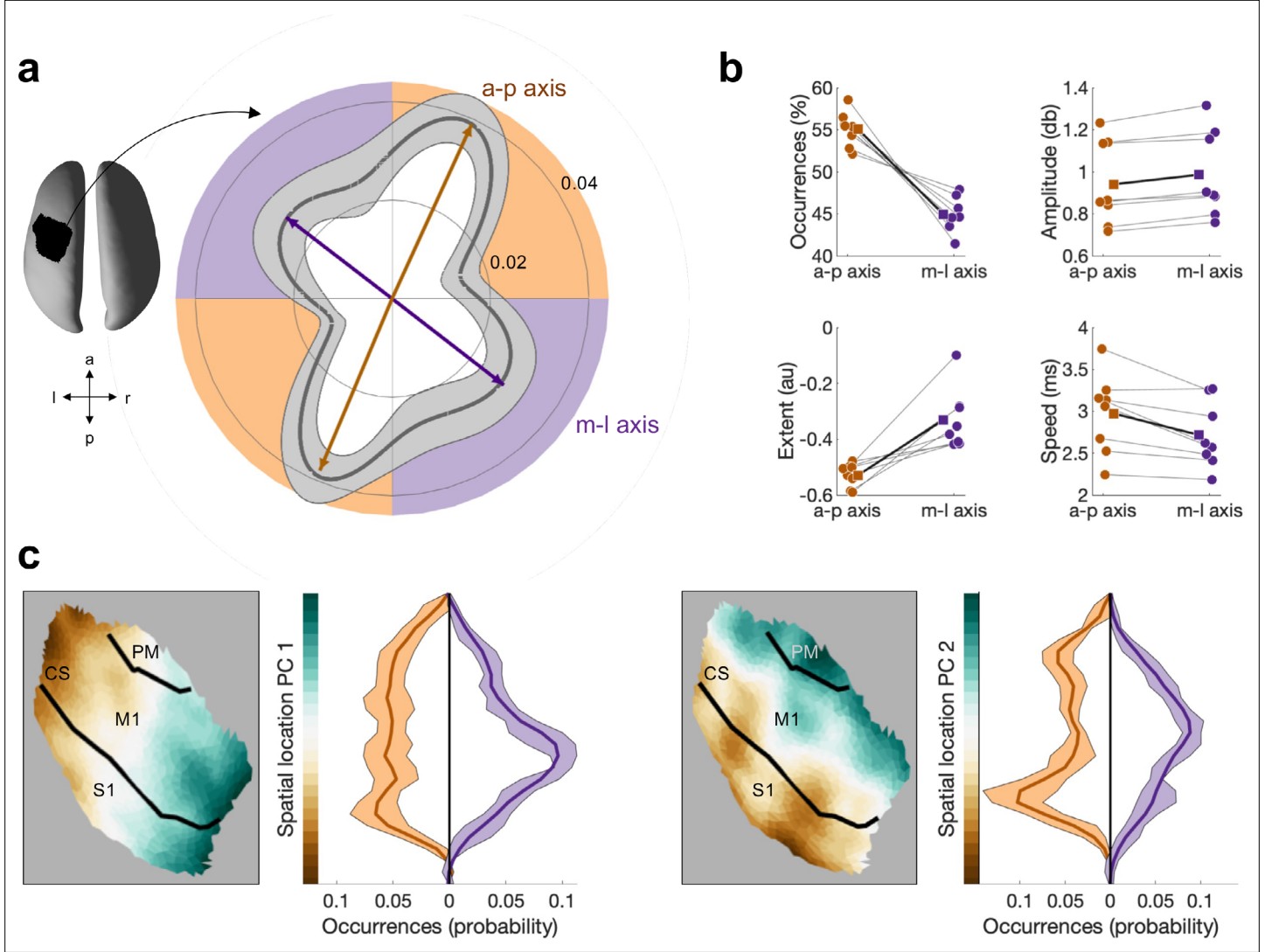

**Figure 4.** Beta bursts activity propagates along one of two axes, which have distinct bursts properties. (**a**) Polar probability histogram showing the probability distribution of burst direction in MNI space. Probability distributions were calculated for each subject individually and then averaged (dark grey line, N=8). Variability across subjects is expressed as standard deviation from the mean (light grey area). To estimate the dominant propagation directions, a mixture of von Mises functions was fitted to the averaged probability distribution (arrows). The four functions lie on two axes. One axis has an anterior–posterior orientation which is approximately perpendicular to the orientation of the central sulcus (a-p), while the other axis runs in approximately medial–lateral orientation which is approximately parallel to the orientation of the central sulcus (m-l). (**b**) Burst occurrence, burst amplitude, burst extent, and burst speed differ as a function of propagation direction. Medians are shown for each subject (circles) and the mean across the subjects' medians (squares). (**c**) Burst location differs as a function of burst direction. Burst location is described by two principal components (PCs) of the Cartesian coordinates of the centre of the burst. For each of the two PCs, the surface plot of the component structure and the probability distributions of the PC score are shown. Probability distributions were calculated for each subject individually and then averaged (dark line, N=8). Variability across subjects is expressed as standard deviation from the mean (light area). Bursts with a direction parallel to the CS, relative to bursts with a direction perpendicular to the CS, are located more centrally in the region of interest (ROI). CS, central sulcus; S1, primary sensory cortex; M1, primary motor cortex; PM, premotor cortex.

The online version of this article includes the following figure supplement(s) for figure 4:

**Figure supplement 1.** Length and angle are highly replicable for all four von Mises functions across halves of the data.

**Figure supplement 2.** Reducing temporal duration, frequency spread, and apparent spatial width to burst extend using principal component analysis (PCA).

**Figure supplement 3.** The spatial location of a burst can be summarised by the first two principal components (PCs) of the Cartesian coordinates of the centre of the burst.

**Figure supplement 4.** Burst frequency and burst location are comparable for pre-movement and post-movement bursts.

axes were well described by a mixture of four von Mises functions (log-likelihood = −1.8058e+04 [For comparison, log-likelihood for two von Mises functions = -1.8231e+04, and log-likelihood for four von Mises functions of a random distribution = -1.8462e+04.]) with means of 66° and 248° for the a-p axis, and means of 142° and 324° for the m-l axis, indicating that the surface mesh imposes structure. Note, however, that these axes do not align with the directions showing the smallest or the largest error when estimating the direction from noise-free gradients on the same surface mesh and spatial burst properties, indicating that the mesh properties do not drive the observed propagation direction.

The reliability of von Mises functions was assessed using a split-half reliability test. In total, 500 split halves were computed and four von Mises functions estimated on each half independently (see 'Statistical analysis'). The length and direction of the von Mises functions were highly reproducible for all four von Mises functions across both halves of the data (percentage difference in length: $M$ = 4.32%, SD = 3.86%; angular difference: $M$ = 2.2°, SD = 2.8°; across 500 repetitions and four von Mises functions; *Figure 4—figure supplement 1*). Further, we tested whether the four von Mises functions were significantly different from zero using non-parametric permutation testing. In total, 5000 permutations were carried out by randomising the propagation direction of each burst and estimating four von Mises functions of the distribution of all bursts. The length of the real van Mises functions was significant while correcting for multiple comparison at $p<0.01$.

Next, we performed a set of control analysis. First, to examine whether the two main propagation axes can be trivially explained by spatial variability in the beamformer weights, we correlated the latency of the critical points across space before and after regressing out the main components of the spatial variability in the linearly constrained minimum variance (LCMV) weights (see 'Control analysis'). We found significant correlations (Pearson's $r$: $M$ = 0.61, SD = 0.27 across individuals, all $ps<0.05$), indicating that beamformer weights contribute to, but do not solely explain the observed propagation directions. Second, we examined whether the distribution of propagation direction is biased by a potential bias in the spatial domain (i.e. *Ahlfors et al., 2010*; *Eulitz et al., 1997*). To test this, we conducted the same analysis on a subset of bursts, forming a uniform spatial distribution (see 'Statistical analysis'). We found that the distribution of propagation directions using this subset of bursts is comparable to the original set of bursts, suggesting that potential spatial sampling bias does not bias the propagation analysis.

Together, these results demonstrate that propagation of sensorimotor beta burst activity occurs along two, orthogonal axes which are oriented approximately parallel and perpendicular to the CS.

## Burst characteristics differ as a function of the propagation axis

The aforementioned analyses suggest that burst activity propagates along one of two propagation axes. We next asked whether bursts propagating along these distinct axes vary in their physiological properties. Specifically, we tested for potential differences in the temporal (i.e. temporal centre), spectral (i.e. frequency centre), or spatial domain (spatial location), as well as burst extent, burst amplitude, and propagation speed.

We found significantly more bursts propagating along the a-p axis ($M$ = 55.1%, SD = 2.0 across individuals), compared to the m-l axis ($M$ = 44.9, SD = 2.0 across individuals; $T$ [test statistic for Wilcoxon signed-rank test, see 'Statistical analysis'] = 2.521, $p<0.012$; *Figure 4b*). Moreover, bursts propagating along these axes differ in their amplitude, extent, speed (*Figure 4b*), and spatial location (*Figure 4c*). Specifically, bursts propagating medial–lateral are characterised by a higher burst amplitude (a-p $M$ = 0.94, SD = 0.20 across individuals; m-l: $M$ = 0.98, SD = 0.20 across individuals; $T$ = 2.521, $p<0.012$), larger extent (a-p: $M$ = −0.53, SD = 0.04 across individuals; m-l: $M$ = −0.33, SD = 0.1 across individuals; $T$ = 2.521, $p<0.012$), and slower propagation speeds (a-p: $M$ = 2.97 m/s, SD = 0.47 m/s across individuals; m-l: $M$ = 2.72 m/s, SD = 0.40 m/s across individuals; $T$ = 2.38, $p<0.017$).

The notion that burst activity propagates along distinct anatomical axes was further supported by differences in the spatial location of bursts propagating along these axes. Specifically, the distribution of spatial location principle component (PC)1 and PC2 (see 'Burst characteristics'; *Figure 4—figure supplement 3*) differed significantly for burst propagating along axis m-l, relative to bursts propagating along axis a-p for both PC1 (KS [test statistic for Kolmogorov–Smirnov test, see 'Statistical analysis']: $M$ = 0.182 across individuals, range = 0.107–0.232; 8/8 $ps<0.001$) and PC2 (KS: $M$ = 0.203 across individuals, range = 0.110–0.253; 8/8 $ps<0.001$; *Figure 4c*). This indicates that bursts

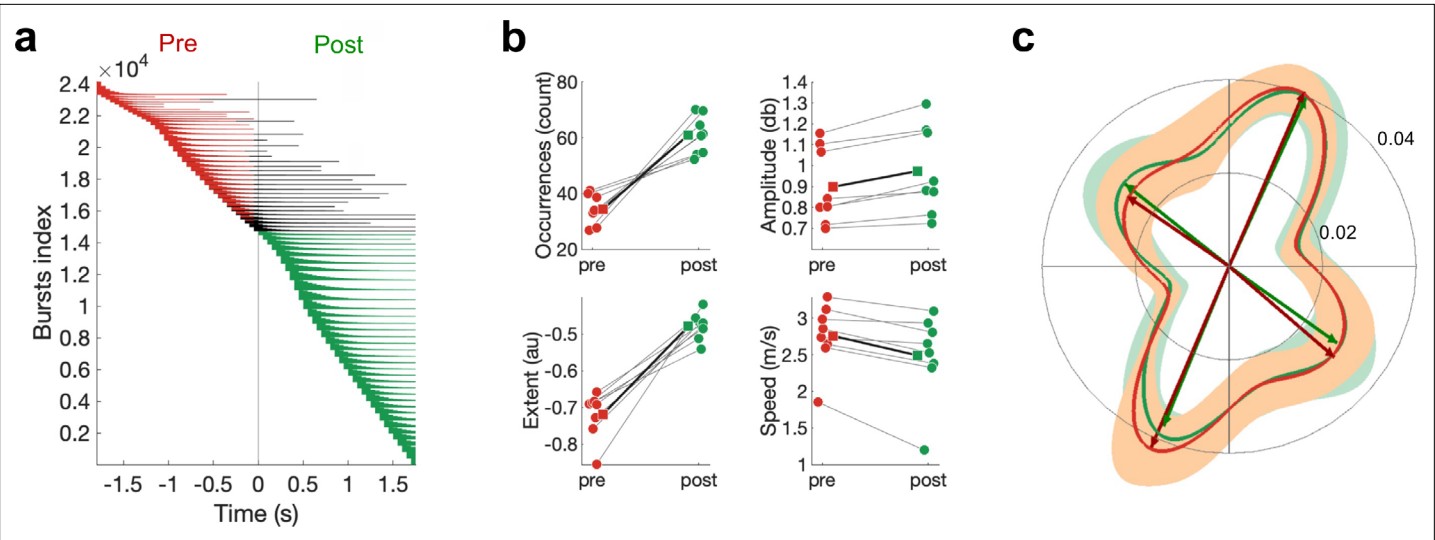

**Figure 5.** Differences in pre- and post-movement beta bursts. (**a**) Burst timing relative to the button press across all subjects. Each horizontal line represents one burst. Bursts are sorted by burst onset and burst duration using multiple-level sorting, yielding the burst index. Pre-movement bursts (i.e. bursts that start and end prior to the button press) are highlighted in red, post-movement bursts (i.e. bursts that start after the button press) are highlighted in green, and bursts that start prior to the button press and end after the button press are highlighted in black. (**b**) Number of bursts, burst amplitude, burst extent, and burst speed differ between pre- and post-movement bursts. Medians are shown for each subject (circles) and the mean across the subjects' medians (squares). (**c**) Propagation direction does not differ between pre- and post-movement bursts. Shown are polar probability histograms separately for pre- and post-movement bursts. Probability distributions were calculated for each subject individually and then averaged (dark line, N=8). Variability across subjects is expressed as standard deviation from the mean (light area). Von Mises functions were fitted separately for pre- and post-movement bursts.

propagating along a-p are located predominantly in the putative hand region of M1 in the vicinity of the central sulcus, whereas the central locus of burst activity propagating mediolaterally is in S1.

## Distinct physiological fingerprints of pre- and post-movement bursts

Having established that sensorimotor burst activity propagates along one of two major axes, with distinct foci of burst activation for burst activity propagating along these, we turned to the question whether bursts occurring pre- or post-movement might also be distinguished by their burst and/or propagation properties. To this end, we defined pre-movement bursts as bursts with an on- and offset prior to the movement, and post-movement bursts as bursts with an on- and offset post movement (*Figure 5a*). Bursts with an onset pre-movement and offset post-movement ($M$ = 4.7%, SD = 1.9% across individuals) are excluded from this specific analysis. As expected, we found significantly more bursts post- than pre-movement (pre: $M$ = 34.3%, SD = 5.3% across individuals; post: $M$ = 60.9, SD = 6.9% across individuals; $T$ = 2.521, p=0.012; *Figure 5b*). Further, post-movement bursts are characterised by a larger amplitude (pre: $M$ = 0.898 db, SD = 0.181 db across individuals; post: $M$ = 0.974 db, SD = 0.208 db across individuals; $T$ = 2.521, p=0.012; *Figure 5b*) and were generally larger in all signal dimensions (burst extent; pre: $M$ = –0.719, SD = 0.063 across individuals; post: $M$ = –0.480, SD = 0.036 across individuals; $T$ = 2.521, p=0.012; *Figure 5b*). However, the average spatial location and frequency centre were not significantly different between pre- and post-movement bursts (all ps>0.1; *Figure 4—figure supplement 4*).

Further, in line with non-human primate recordings (*Rubino et al., 2006*), propagation directions were not significantly different between pre- and post-movement bursts ($U^2$ [test statistic for Watson's $U^2$ test, see 'Statistical analysis']: $M$ = 0.088 across individuals, range = 0.025–0.190; 8/8 ps>0.1; *Figure 5c*). The directions of pre-movement bursts activity propagating along the a-p (68/246°) and m-l direction (148/315°) did not differ from the directions observed post-movement (a-p: 66/248°; m-l: 142/325°). However, while the mean propagation direction did not differ between pre- and post-movement bursts, we found that propagation speed for post-movement bursts was significantly slower than pre-movement (pre: $M$ = 2.90 m/s, SD = 0.20 m/s across individuals; post: $M$ = 2.69, SD = 0.28 across individuals; $T$ = 2.521, p=0.012; *Figure 5b*). Finally, we sought to explore whether

pre-movement burst characteristics are related to reaction time. We did not find evidence that burst characteristics relate to reaction time in these data (all ps>0.1).

## Discussion

The temporal, spectral, and spatial characteristics of beta bursts in human sensorimotor cortex remain unknown. We here show that beta bursts in human sensorimotor cortex occur predominantly post-movement, in the lower beta frequency band, and on the posterior bank of the precentral gyrus. Crucially, sensorimotor beta bursts do not just occur as local standing waves of synchronous activity but propagate along one of two axes that run parallel or perpendicular to the central sulcus, respectively. In addition to the principal axis of their propagation direction, these bursts differ in their occurrence, location, propagation speed, amplitude, and extent. Further, post-movement bursts are characterised by higher amplitude, larger extent, and slower propagation speed, suggesting distinct physiological markers and functional roles pre- and post-movement. Collectively, our data provide novel evidence that a substantial proportion of human sensorimotor beta burst activity travels along two anatomical and functional distinct axes, with distinct burst properties pre- and post-movement.

### Distinct anatomical propagation axes of sensorimotor beta activity

Travelling wave activity occurs at multiple spatial scales, ranging from mesoscopic, columnar, to macroscopic, transcortical levels (*Muller et al., 2018*). Here we show that beta burst activity in human sensorimotor cortex propagates along one of two approximately orthogonal axes that are oriented in anterior–posterior and medial–lateral direction. Recordings from invasive multi-electrode arrays confirm the dominance of these two propagation axes, albeit on a different spatial scale. For example, Takahashi and colleagues reported that beta activity in M1 of a tetraplegic patient propagated along the medial–lateral axis (*Takahashi et al., 2011*). And using a multi-thousand channel array placed across the central sulcus beta activity propagated along the anterior–posterior axis across the M1-S1 functional boundary (*Tchoe et al., 2022*). In non-human primates, beta activity propagates along the anterior–posterior axis in M1 (*Balasubramanian et al., 2020*; *Best et al., 2017*; *Rubino et al., 2006*; *Takahashi et al., 2011*; *Takahashi et al., 2015*), and along the medial–lateral axis in the dorsal premotor cortex (*Rubino et al., 2006*), indicating regional differences in spatiotemporal patterns (*Rubino et al., 2006*; *Rule et al., 2018*).

Except for the multi-thousand channel array, which can cover an area of roughly 64 cm$^2$ in humans (*Tchoe et al., 2022*), neural activity has been recorded from a single cortical region, limited by the dimension of the electrode array (roughly 0.16 cm$^2$). By contrast, we here identified spatiotemporal patterns of beta activity in burst events of an average apparent spatial width of ~6 cm$^2$ located in M1 and adjacent cortical areas. By leveraging high SNR MEG recordings that permit high sensitivity in all signal domains, we were able to quantify bursts and their spatiotemporal pattern non-invasively over these functionally cogent brain regions at a spatial scale that sits between invasive recordings in animals and previous human M/EEG or intracranial recordings (*Alexander et al., 2016*; *Roberts et al., 2019*; *Rule et al., 2018*; *Stolk et al., 2019*; *Takahashi et al., 2011*). Our results extend previous invasive recordings by showing that bursts activity can travel across sensory and motor cortices and bridge across functionally distinct brain areas.

The spatial profiles of propagation of beta activity along the anterior–posterior and medial–lateral direction are in line with the idea that propagation directions are imposed by the dominant internal connections within anatomical networks (*Rubino et al., 2006*). Here, our dominant propagation axes conformed to an anterior–posterior network comprising dorsal premotor cortex, primary motor cortex, and primary sensory cortex (*Cauller et al., 1998*; *Kurata, 1991*; *Luppino and Rizzolatti, 2000*; *Muakkassa and Strick, 1979*) and a medial–lateral network spanning across medial and lateral dorsal premotor cortex, supplementary motor area cortex, and caudal portions of ventral premotor cortex (*Dum and Strick, 2005*; *Ghosh and Gattera, 1995*; *Luppino et al., 1993*). The latter is thought to mirror proximal and distal sites within the motor cortex (*Rubino et al., 2006*), with proximal representations (i.e. shoulder and elbow) located more medially and distal representations (i.e. wrist and fingers) located more laterally (*Penfield and Boldrey, 1937*). This suggests that at a macro-scale level, the direction of wave propagation is dictated by the underlying horizontal connections, though further work across different spatial scales (such as

*Sreekumar et al., 2020*) is required to fully unpack the precise relationship between sustained rhythmic synchronous spiking activity within neural populations, mesoscopic and macroscopic traveling wave activity.

While our results further corroborate the importance of anterior–posterior and medial–lateral propagation axes, the precise mechanism of travelling wave activity remains unclear. One possible mechanism is that excitation from a single generator propagates through a network, guided by conduction delays within corticocortical and the corticothalamic system (*Ermentrout and Kleinfeld, 2001*; *Muller et al., 2018*; *Prechtl et al., 2000*). Alternatively, travelling wave activity could arise from one generator driving a network through increasing time delay, so-called fictive travelling waves, or coupled generators that exhibit stable phase differences (e.g. *Zhigalov and Jensen, 2022*). Different levels of network interactions may thus generate and sustain propagating waves. Common to all travelling wave activity is the idea that they generate a consistent spatiotemporal frame for further neuronal interactions. In mesoscopic data, it is very challenging to analytically resolve any ambiguity about the mechanism of wave generation. LFP recordings with implanted electrode arrays in non-human primates suggest that coupled oscillators contribute significantly to beta travelling waves over a spatial scale of 0.16 cm$^2$ (*Rule et al., 2018*).

## Propagation axes of sensorimotor beta activity are physiologically distinct

While previous work has investigated individual aspects of neural activity in relation to propagation direction (*Balasubramanian et al., 2020*; *Bhattacharya et al., 2022*), we here consider all signal domains of neural activity. We found that the two propagation axes can be distinguished based on their physiological properties, such as propagation speed, burst occurrence, amplitude, and extent. Specifically, beta activity propagating in the medial–lateral direction is characterised by higher burst amplitude and larger burst extent, that is, bursts are larger in all signal domains. Further, more bursts propagate along the anterior–posterior l direction, which is also characterised by faster propagation speed.

Propagating wave activity can occur in a wide range of different speeds, with propagation speeds broadly falling into two categories. Speeds for mesoscopic travelling waves occurring within cortical columns and their lateral connections, as identified using local field potential (LFP), multielectrode arrays, or optical imaging, range between 0.1 and 0.8 m/s (*Bhattacharya et al., 2022*; *Rubino et al., 2006*; *Takahashi et al., 2011*; *Takahashi et al., 2015*). These slower wave speeds are consistent with axonal conduction speeds of unmyelinated horizontal fibres in the superficial layers of the cortex (*Girard et al., 2001*).

By contrast, macroscopic travelling waves spanning across several cortical regions, and commonly assessed using mass-neural signal recordings such as M/EEG or ECoG, have been reported at speeds ranging from around 1 to 10 m/s (*Alexander et al., 2016*; *Hughes, 1995*; *Muller et al., 2018*). The relatively large variability in propagation speed of macroscopic travelling waves is partly due to variability in spatial resolution with low spatial resolution being susceptible to aliasing artefacts (*Alexander et al., 2016*; *Bahramisharif et al., 2013*), and some uncertainty in the travelled distance. Regarding the latter, while it has been recommended to study travelling waves on the cortical surface (*Alexander et al., 2019*; *Hughes, 1995*) it is still unclear whether neural activity truly propagates along the brains cortical surface (as quantified by geodesic distance), or, at least in part, propagate through the brain volume (as quantified by Euclidean distance). Further, propagation distance can be computed on the original, folded cortical surface or on the inflated surface. Our data show that propagation speed derived from the folded surfaces is roughly twice as fast than the propagation speed derived from the inflated surface (*Figure 2—figure supplement 2*), which is in line with the previously reported folding factor of x2.2 (*Alexander et al., 2016*; *Burkitt et al., 2000*). Notwithstanding the uncertainty this introduces in estimating propagation speeds, the range of propagation speeds observed here are compatible with previous reports from human and non-human primates (*Hughes, 1995*; *Muller et al., 2018*), and are compatible with axonal conduction speeds of myelinated cortical white matter fibres (*Swadlow and Waxman, 2012*), suggesting an active role for macro-scale travelling burst activity in intra-areal communication and information transfer.

## Pre- and post-movement burst are expressed differently

The transient bursts of beta activity in our human MEG data lasted, on average, several hundred milliseconds, and span over approximately 3 Hz predominantly in the lower beta frequency range. These temporal and spectral properties are broadly in line with previous reports (*Cagnan et al., 2019*; *Quinn et al., 2019*; *Seedat et al., 2020*; *Shin et al., 2017*; *Sporn et al., 2020*; *Tinkhauser et al., 2017a*), with variation in the absolute values being strongly dependent on how bursts are operationalised (*Zich et al., 2020*). We extend these previous reports on the temporal and spectral burst characteristics by additionally characterising spatial burst characteristics. Sensorimotor beta burst activity often spans over several square centimetres with a distinct topographic distribution. The majority of bursts are located on the posterior bank of the precentral gyrus, with a proportion of bursts that spread to adjacent areas. While approaching the spatial limits of human MEG, these data indicate the possibility of locating beta activity within the sensorimotor cortex.

To further elucidate the functional roles of sensorimotor beta bursts, we next compared pre- and post-movement bursts with regard to both their temporal, spectral, and spatial burst characteristics, and their propagation properties. We confirmed that post-movement, compared to pre-movement, bursts occur more frequently and are stronger (i.e. higher burst amplitude) and larger in all signal domains (i.e. larger burst extent). These observations are largely in line with previous studies (*Quinn et al., 2019*; *Seedat et al., 2020*; *Zich et al., 2018*), whereby we note that (*Little et al., 2019*) no difference in temporal burst duration between pre- and post-movement bursts was reported. We believe this discrepancy is because (*Little et al., 2019*) employed different thresholds for pre- and post-movement bursts, whereby here the same threshold was used. Moreover, we find that pre-movement bursts exhibit faster propagation speed than post-movement burst activity. There is no evidence that the difference in propagation speed is mediated through differences in the frequency (*Alexander et al., 2016*), or spatial location of bursts, as both metrics are comparable for pre- and post-movement bursts. The functional relevance of this difference in propagation speed merits further consideration in the future, but it indicates that parsing the functional role of beta activity may require its decomposition into its physiologically distinct stationary and propagating components. Finally, we show that pre- and post-movement bursts propagate along the same propagation axis, which is in line with previous reports, observing the same propagation axes during action (*Rubino et al., 2006*) and rest (*Takahashi et al., 2011*). This provides further evidence that the propagation of burst activity is constrained by the underlying connectivity.

Together, our results show that, compared to pre-movement bursts, post-movement bursts are stronger and larger in all signal domains, whereby their spectral and spatial centre, as well as their propagation direction, are comparable. This might indicate that pre- and post-movement bursts share the similar generator processes, which exhibits more and stronger bursts post-movement. Studies using biophysical modelling have proposed that beta bursts are generated by a broad infragranular excitatory synaptic drive temporally aligned with a strong supragranular synaptic drive (*Law et al., 2022*; *Neymotin et al., 2020*; *Sherman et al., 2016*; *Shin et al., 2017*) whereby layer-specific inhibition acts to stabilise beta bursts in the temporal domain (*West et al., 2023*). The supragranular drive is thought to originate in the thalamus (*Jones, 1998*; *Jones, 2001*; *Mo and Sherman, 2019*; *Seedat et al., 2020*), indicating thalamocortical mechanisms. Interestingly, sensorimotor beta bursts have not only been observed during action but also during rest (*Zich et al., 2018*; *Seedat et al., 2020*; *Becker et al., 2020*; *Echeverria-Altuna et al., 2021*), which raises the question of their functional role. That sensorimotor beta bursts occur across functional states, spatial scales, and species suggests that the functional role of the mere presence of bursts is a very elementary one, such as maintaining the 'status quo' (*Engel and Fries, 2010*) or 'null space' (*Kaufman et al., 2014*). In addition, we believe that specific functional roles can be linked to the manifestation of bursts quantifiable by their temporal, spectral, and spatial bursts characteristics as well as their propagation properties. To give one example, motor symptoms in Parkinson's disease have been linked to prolonged burst duration (*Deffains et al., 2018*; *Tinkhauser et al., 2017a*; *Tinkhauser et al., 2017b*). The proposed hierarchical dual-role framework of burst function can be tested using biophysical models (*Neymotin et al., 2020*) and targeted neuromodulatory experiments.

## Caveats of spatial and spatiotemporal properties in source space

Non-invasive techniques have limitations that should be considered when interpreting the spatial domain of bursts and travelling wave activity. LCMV beamformers assume that each source is a single dipole and that there are no other correlated sources in the brain. These limitations make interpretation of spatial structure in LCMV power maps ambiguous. We explore several specific issues: firstly, whether the apparent spatial extent of a source is simply modulated by the SNR of the signal. Secondly, the inherent smoothness of the source reconstruction maps due to the mapping of a few hundred sensors to several thousand voxels. Finally, if a patch of cortex is active rather than a single point source, then these correlated voxels can suppress the signal of interest. Each of these points can be challenging when interpreting the spatial domain of bursts and travelling waves.

The first issue suggests that differences in the bursts' apparent spatial width could simply be caused by differences in SNR across and/or within sessions rather than differences in the spatial distribution of cortical activity. We performed one beamformer per session; thus, different SNR levels across sessions would affect the beamformer weights. However, if variation across sessions in beamformer weights would explain variation in bursts' apparent spatial width, we would expect a negative relationship between burst amplitude and burst apparent spatial width across sessions. This is not the case in our data, suggesting that between-session differences in beamformer weights do not cause the observed differences in bursts' apparent spatial width. Nevertheless, spatial width of burst activity measured with M/EEG or ECoG should be interpreted with caution. Here, due to the strong correlation between the bursts' apparent spatial width, temporal duration, and frequency spread, we combined these signal properties using PCA and used the resulting cross-modal measure burst extent.

Secondly, the inherent smoothness of the beamformer solution can lead to 'trivial' structure in the source solution, meaning that single sources can leak across cortex or that multiple sources can become blurred together. Across space in bursts diverse phase lags exists, suggesting that structure is unlikely to have arisen solely from leakage of a single source. The functional role of travelling waves remains unclear. As outlined above, the mechanisms underlying travelling waves remain ambiguous (see 'Distinct anatomical propagation axes of sensorimotor beta activity'), both at the meso- and macro scale (*Hughes, 1995*; *Muller et al., 2018*). We cannot rule out the possibility that this phase structure arises from mixing of multiple distinct sources but take a 'gradient' or 'travelling wave' perspective here to better link with comparative literature. While this concerns travelling wave analyses across a range of spatial scales and recording techniques, source space analysis, as employed here, entails an additional issue – namely whether the propagation directions can be trivially explained by spatial variability in the beamformer weights. Our control analysis showed that the estimated propagation direction correlates significantly with the propagation direction obtained after regressing out the main components of spatial variability in the beamformer weights. This indicates that beamformer weights can contribute to, but do not solely explain, spatiotemporal gradients in human MEG data.

Finally, patches of high-amplitude, correlated sources can be mutually suppressed by the LCMV beamformer, leading to an apparent loss of signal. Though we cannot remove the possibility these mutual correlations may be suppressing part of the signal, we observe strong task-related activity suggesting that a substantial proportion remains in our analysis.

Together, we acknowledge that the beamformer weights can affect bursts' spatial width and propagation direction but believe that our control analyses suggest that the beamformer weights are not driving the observed effects.

## Materials and methods

### Participants and experimental task

The study was approved by the UCL Research Ethics Committee (reference number 5833/001) and conducted in accordance with the Declaration of Helsinki. Informed written consent was obtained from all participants. All participants (six males, *M* = 28.5 y, SD = 8.52 y across individuals) were free of neurological or psychiatric disorders, right-handed, and had normal or corrected-to-normal vision.

Participants performed a visually cued action decision-making task in which they responded to visual stimuli projected onto a screen by pressing one of two buttons using their right index or middle finger (for details, see *Bonaiuto et al., 2018*). Each trial consists of a baseline (1–2 s), random dot kinematogram (RDK, coherent motion to the left or right for 2 s), delay period (0.5 s), instruction

cue (arrow pointing to the left or right), and motor response. Participants were instructed to press the corresponding button (index finger for left button and middle finger for right button) as fast and accurately as possible. The task uses a factorial design with congruence between RDK and cue (congruent, incongruent) and coherence of the dot motion (low, medium, high). Each block comprised 126 congruent and 54 incongruent trials, and 60 trials for each coherence level with half containing leftward motion, and half rightward motion. Here we only consider congruent, high-coherence trials (42 trials per block) that were responded to correctly.

## MRI acquisition and processing

Prior to the MEG sessions, structural MRI data were acquired using a 3T Magnetom TIM Trio MRI scanner (Siemens Healthcare, Erlangen, Germany). A T1-weighted 3D spoiled fast low-angle shot (FLASH) sequence was acquired to generate an accurate image of the scalp for head-cast construction. Subject-specific head-casts optimise co-registration and reduce head movements, and thereby significantly improve the SNR. See *Bonaiuto et al., 2018*; *Meyer et al., 2017*; *Troebinger et al., 2014* for details on the sequence and the head-cast construction.

In addition, a high-resolution, quantitative, multiple parameter mapping (MPM) protocol, consisting of three differentially weighted, RF and gradient spoiled, multi-echo 3D FLASH acquisitions recorded with whole-brain coverage at 800 mm isotropic resolution, was performed. See *Bonaiuto et al., 2018*

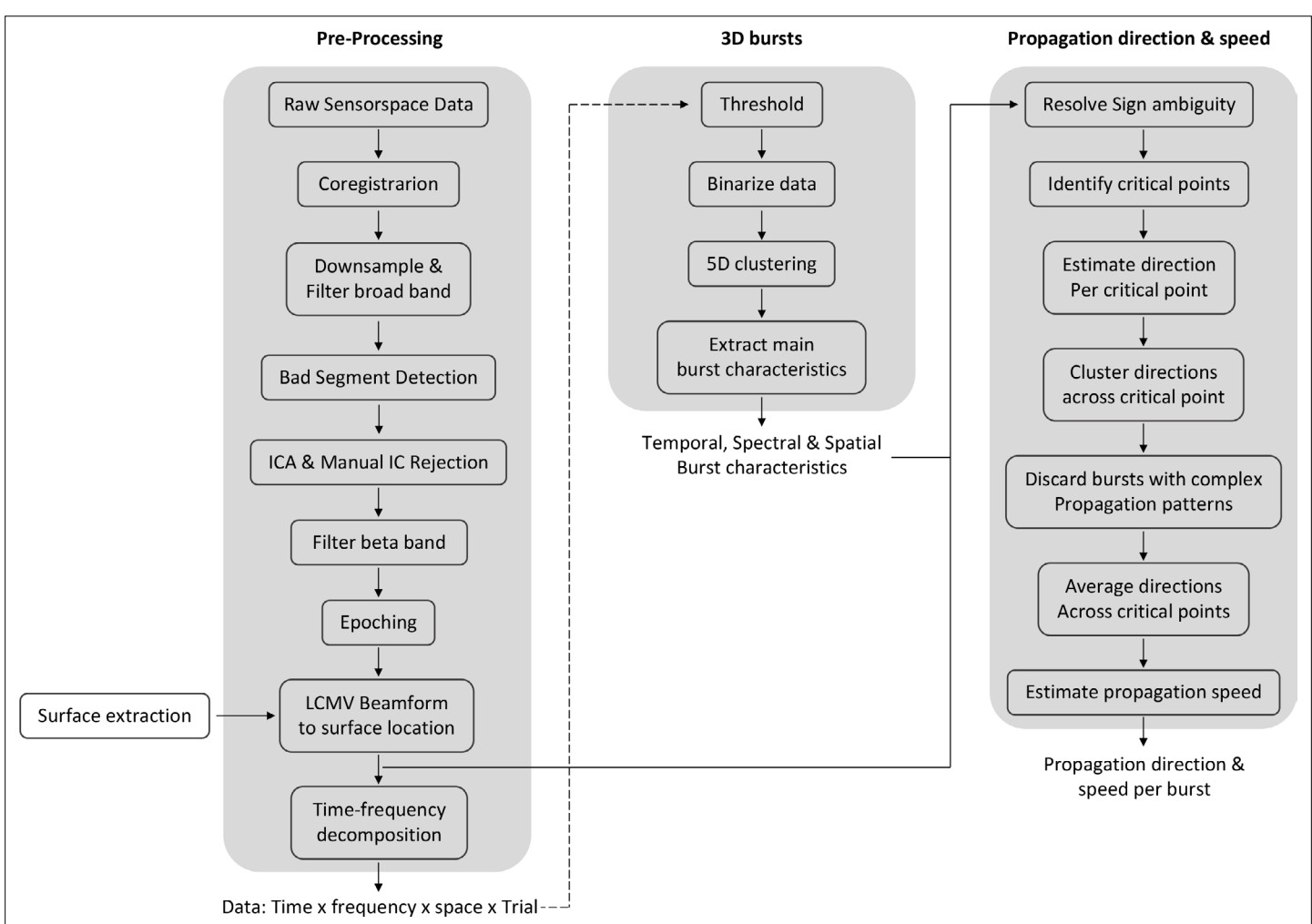

**Figure 6.** A schematic for the processing pipeline.

The online version of this article includes the following figure supplement(s) for figure 6:

**Figure supplement 1.** Empirical threshold to binarise beta bursts.

**Figure supplement 2.** Effect of different burst thresholds on burst characteristics, propagation speed and direction.

for details on the protocol. Each quantitative map was co-registered to the scan used to design the head-cast using the T1 weighted map. Individual cortical surface meshes were extracted using Free-Surfer (v5.3.0; *Fischl, 2012*) from multiparameter maps using the PD and T1 sequences as inputs, with custom modifications to avoid tissue boundary segmentation failures (*Carey et al., 2018*). Meshes were down-sampled by a factor of 10 (vertices: $M$ = 30,095, SD = 2665 across individuals; faces: $M$ = 60,182, SD = 5331 across individuals) and smoothed (5 mm). Here we used the original and the inflated pial surface.

## MEG acquisition and pre-processing

MEG data were acquired using a 275-channel Canadian Thin Films (CTF) MEG system using individual head-casts in a magnetically shielded room. Head position was localised using three fiducial coils placed at the nasion and left/right pre-auricular points, within the head-cast. Data were sampled at 1200 Hz. A projector displayed the visual stimuli on a screen (~50 cm from the participant), and participants made responses with a button box.

A summary of the data processing pipeline is shown in *Figure 6*. MEG data were processed in for each block separately unless stated otherwise. Firstly, raw data were converted to SPM12 format for analysis in Matlab2019b. Registration between structural MRI and the MEG data was performed with RHINO (Registration of head shapes Including Nose in OSL) using only the Fiducial landmarks and single shell as forward model. Unless stated otherwise, data were analysed in single subject space.

Continuous data were down-sampled to 300 Hz. Further, a band-pass (1–95 Hz) and notch-filter (49–51 Hz) were applied. Time segments containing artefacts were identified by using generalised extreme studentised deviate method (GESD; *Rosner, 1983*) on the standard deviation of the signal across all sensors in 1 s non-overlapping windows, with a maximum number of outliers limited to 20% of the data and adopting a significance level of 0.05. Data segments identified as outliers were excluded from subsequent analyses.

Further, denoising was applied using independent component analysis (ICA) using temporal FastICA across sensors (*Hyvärinen, 1999*). Sixty-two independent components were estimated, and components representing stereotypical artefacts such as eye blinks, eye movements, and electrical heartbeat activity were manually identified and regressed out of the data.

Data were then filtered to the frequency band of interest (β 13–30 Hz) and segmented from –2 s to 2 s relative to the button press. Segmented data were projected onto subjects' individual cortical surface meshes using an LCMV vector beamformer (*Van Veen and Buckley, 1988*; *Woolrich et al., 2011*). The beamformer weights were estimated at the centre of each face, referred to henceforth as spatial locations. A covariance matrix was computed across all segments and was regularised to 55 dimensions using principal component analysis (PCA). All analyses are conducted in source space.

## Time-frequency decomposition

Time-frequency analysis was applied to single trials and spatial locations using dpss-based multitaper (window = 1.6 s, steps = 200 ms) with a frequency resolution of 1 Hz. Epochs were baseline corrected (–1.8 s to –1.1 s). This procedure results in a trial-by-trial time-frequency decomposition for each spatial location, that is, relative power in 4D, time × frequency × space × trial, whereby space is on its own three-dimensional (x, y, z coordinates of surface locations).

## Burst operationalisation

We used binarisation and high-dimensional clustering to operationalise beta bursts. Power derived from time-frequency analysis was first binarised using a simple amplitude threshold. The threshold was obtained empirically as in previous work (*Little et al., 2019*). Specifically, trial-wise power was correlated with the burst probability across a range of different threshold values (median to median plus 7 standard deviations, in steps of 0.25). The threshold value that retained the highest correlation between trial-wise power and burst probability was used to binarise the data. To account for difference in signal-to-noise across sessions, days and subjects we obtained one threshold per session ($M$ = 2.97 × SDs above mean, SD = 0.66 across sessions; *Figure 6—figure supplement 1*). To explore the robustness of the results, analyses were repeated using a range of thresholds (*Figure 6—figure supplement 2*).

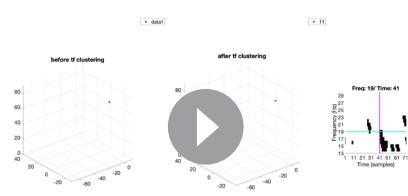

**Video 1.** Illustration of 5D clustering.
https://elifesciences.org/articles/80160/figures#video1

Detecting bursts simultaneously in the temporal, spectral, and spatial domain is accompanied by some conceptual and computational challenges. Here we opt for a simple thresholding approach rather than a more data-driven approach, such as the hidden Markov model (HMM; *Quinn et al., 2019*; *Vidaurre et al., 2016*). Firstly, existing HMM variants do not provide the here desired frequency resolution. Secondly, adapting the amplitude-envelope HMM to threshold power derived from time-frequency analysis poses a computational challenge for this high-dimensional dataset. Finally, one of the main advantages of HMM, that is, the prevention of burst-splits (see *Quinn et al., 2019*) is overcome in the 5D clustering procedure. Together, while HMM and other data-driven approaches are generally advantageous, in this framework a simple amplitude threshold is preferred.

Following binarisation, data were clustered across time, frequency, and space using a three-stage approach (see *Video 1*). Note that data are four-dimensional, that is, time × frequency × space × trial, whereby space is on its own three-dimensional (x, y, z coordinates of surface locations). First, for each trial data were clustered in 2D (i.e. time × frequency). To this end, the binarised data were summed over the spatial domain and time-frequency cells with at least one surface location being 'on' were clustered using eight-connectivity (i.e. connected horizontally, vertically, or diagonally). Second, for each time-frequency cell with at least one surface location being 'on', spatial locations on the surface mesh were clustered in 3D (i.e. x, y, z coordinates of surface locations). Spatial locations were part of the same cluster if their Euclidean distance was smaller than the maximal distance of two spatial locations ($M$ = 2.66 mm; SD = 0.15 mm across individuals). Finally, source clusters were combined across time-frequency cells using eight-connectivity, that is, if two spatial clusters of two adjourning time-frequency cells overlapped in at least one surface location the two spatial clusters were combined. This procedure allows clustering in high-dimensional irregular space and results in 3D (time × frequency × space) bursts. Burst identification was limited to the time of interest (−2 to 2 s relative to the button press), the frequency of interest (13–30 Hz), and ROI (left-hand area). To restrict the burst analysis to an ROI, volume-based ROIs in MNI space were normalised to subject's native space using the inverse deformation field and transformed to surface-based ROIs. Clusters had to span at least two time points, frequency steps, and spatial locations to be considered further.

## Burst characteristics

We divide burst characteristics into first- and second-level burst characteristics. We define first-level burst characteristics as characteristics that are obtained for each burst and each domain separately. *Figure 1* illustrates the first-level characteristics. For the temporal domain, burst temporal on- and offset, temporal duration and centre (i.e. mean of on- and offset) were obtained. Equally, low- and high-frequency boundaries, frequency spread and centre (i.e. mean of low and high boundary) were extracted for the spectral domain. For the spatial domain, we obtained the spatial width (i.e. total surface area defined as the sum of the area of all faces), the size in each dimension (x, y, z) using the minimum bounding rectangle (i.e. bounding box), and the spatial centre. The spatial centre is defined as the projection of the centre of mass onto the surface. The spatial centre can be described using its Cartesian coordinates. An alternative to the description of the spatial centre is provided by the first two components of a PCA of the Cartesian coordinates (*Figure 4—figure supplement 3*). The first two PCs describing 98% of variance are retained for further analysis. The first PC (76.3% variance explained) contains a spatial gradient along the anterior/lateral–posterior/medial axis. The second PC (22.3% variance explained) contains a spatial gradient along the anterior/medial–posterior/lateral axis. Thus, the location of an individual burst can be described by the two PC scores, relating to the amount of each PC that it contains. In addition, burst amplitude was obtained, that is, the mean amplitude across all time points, frequencies, and spatial locations of the burst.

These first-level burst characteristics form the basis for second-level burst characteristics. These can be broadly summarised as (1) combinations and (2) interactions of characteristics within and across

domains. Here we extract one of these measures: temporal duration, frequency spread, and apparent spatial width were combined to a single metric, that is, burst extent. The three measures are highly correlated within subjects ($M = 0.785$; SD = 0.021, across the three correlations and eight individuals, *Figure 4—figure supplement 2a and b*) and were therefore reduced to a single metric using PCA. The first principal component explains 85.6% of the variance and is defined as burst extent (PC 2: 7.5%; PC 3: 6.9%, *Figure 4—figure supplement 2c*). These burst metrics are not exhaustive. Other second-level burst characteristics include the interaction between frequency centre and spatial location (*Zich et al., 2020*), waveform variability (*Szul et al., 2022*), and the variability of the first-level burst characteristics, such as the variability in (instantaneous) frequency, within and across bursts. Future work may address the variability and temporal stability of bursts, for example, using methods such as empirical mode decomposition (*Huang et al., 1998*).

## Propagation direction and speed of neural activity within bursts

To investigate whether activity within human sensorimotor bursts propagates, we identified the dominant propagation direction and speed for each burst. To this end, data (before time-frequency decomposition, see *Figure 6*) of each burst were extracted from burst on- to offset for each surface location in the burst. The sign ambiguity in the beamforming process entails that the spatial locations within a burst may have arbitrarily opposite signs. This is not an issue when estimating power, as above, but can impact on the estimation of the propagation direction. Sign ambiguity was resolved using the sign-flipping algorithm described in *Vidaurre et al., 2018*. For a finer temporal resolution, data were interpolated by a factor of 10.

For each burst, we estimated the propagation direction and propagation speed. Propagation direction and speed were estimated from critical points in the oscillatory cycle (four critical points per oscillatory cycle, i.e. peak and trough as well as peak-trough and trough-peak midpoint, grey vertical lines in *Figure 2a*) and then averaged across critical points within one burst.

The propagation direction at each critical point was estimated from the relative latency (i.e. absolute latency of that critical point at each surface location relative to absolute latency of that critical point for the average across all surface locations in that burst). For example, *Figure 2bi* shows the relative latency for each surface location in the burst for the critical point at 212 ms in the burst. Next, from these relative latencies and their surface locations the propagation direction was estimated. Specifically, propagation direction was estimated using linear regression (*Balasubramanian et al., 2020*), whereby the relative latency at the surface location was predicted from the coordinates of the surface location of the inflated surface. On the inflated surface, a gradient in the z-direction is always depicted by a gradient in x- or y-direction, which is why only two simple linear regressions were estimated, one for the x-and one for the y-direction (*Figure 2bii*). Propagation direction along the x-y-direction was obtained by transforming the regression coefficients from Cartesian coordinates to spherical coordinates (red arrow in *Figure 2biii*). For each regression, its associated coefficient of determination ($R^2$) was calculated and the two $R^2$s averaged. This approach results in one propagation direction and one $R^2$ per critical point.

Propagation direction across critical points was obtained by clustering (i.e. spectral clustering) the propagation directions across critical points. Three scenarios existed: (1) one cluster was obtained and the variability across directions of critical points was relatively low (standard deviation $< \pi/4$; *Figure 2biii*; *Figure 2—figure supplement 1a*); (2) one cluster was obtained and the variability across directions of critical points was relatively high (standard deviation $> \pi/4$; *Figure 2—figure supplement 1b*); and (3) more than one cluster was obtained (*Figure 2—figure supplement 1c*). Scenarios 2 and 3 indicate complex propagation patterns, such as random or circular patterns (*Denker et al., 2018*; *Rule et al., 2018*). Based on previous literature, we expect planar traveling waves to be dominant in the primary motor cortex (*Balasubramanian et al., 2020*; *Rubino et al., 2006*; *Rule et al., 2018*; *Takahashi et al., 2011*). For bursts of scenario 1, propagation directions and $R^2$ were averaged across critical points (back arrow in *Figure 2biii*). To have sufficient confidence in the direction, bursts with an average $R^2 < 0.2$ were discarded (*Balasubramanian et al., 2020*). Following this procedure, we found that 79.59% (SD = 2.37% across individuals) of the bursts show a spatiotemporal pattern ($R^2$: $M = 0.355$, SD = 0.016 across individuals). To combine propagation directions across subjects, propagation directions were spatially normalised to MNI space using the deformation field. Directions are presented as probability distributions. On the average of the

probability distributions across subjects, the propagation direction was quantified using a mixture of von Mises functions.

The propagation speed at each critical point was defined as the distance between the spatial locations with the largest and smallest relative latency (i.e. latency of each surface location relative to the average latency) divided by the difference in their latencies (*Bahramisharif et al., 2013*). Distance was computed using exact geodesic distance (*Mitchell et al., 1987*; *Figure 2cii*) on the inflated surface. Propagation speed was averaged across critical points. For comparison, we calculated the propagation speed using the original surface. The speed computed using the distance on the original surface ($M$ = 4.90 m/s, SD = 0.46 m/s across individuals) is faster than the speed computed using the distance on the inflated surface ($M$ = 2.61 m/s, SD = 0.39 m/s across individuals, *Figure 2—figure supplement 2*). This difference is well in line with a suggested cortical folding factor of x2.2 to adjust propagation speeds for cortical folding (*Alexander et al., 2016*; *Burkitt et al., 2000*). Propagation speed is in the expected range of macroscopic waves (*Hughes, 1995*; *Muller et al., 2018*).

## Accuracy of the propagation direction detection in simulated and real meshes

Using simulation, we evaluated the accuracy of the propagation direction estimation. To this end, we generated 360 noise-free high-resolution gradients span 1° in steps of 1° (*Figure 3a* shows a subset). To evaluate the effect of mesh type and spatial sampling, we created three 2D mesh types, (1) square mesh (*Figure 3b*), (2) circular mesh (*Figure 3c*), and (3) random mesh (*Figure 3d*), whereby each mesh type was sampled at three spatial sampling rates: N/2, N, and N × 2 (N approximates the spatial sampling of the surface mesh, i.e. roughly 27 surface locations per $cm^2$). For each gradient and each mesh, the propagation direction was estimated and the estimation error, that is, difference between true and estimated propagation direction, computed. For the random mesh, this procedure was repeated 100 times, each time with a different random mesh.

As the surface mesh is irregular and each burst is unique in its spatial size and shape, we additionally evaluated the accuracy of the propagation direction estimation for the real bursts. To this end, for each individual burst and each gradient, the propagation direction was estimated, and the estimation error computed as above.

## Control analysis

The ill-posed nature of the inverse problem in M/EEG means that the source estimation has a degree of smoothness. While this is unavoidable and shared with all inverse problem methods, the smoothness can be problematic when interpreting the spatial domain of burst and their spatiotemporal gradients, travelling waves. We perform a series of control analysis to explore the practical effect of these ambiguities in our data. Our reasoning was that with regards to interpreting the spatial width of burst activity, any differences could be caused by differences in SNR across and/or within sessions rather than differences in the spatial distribution of cortical activity (see *Figure 1—figure supplement 2ai and bi* for a schematic illustrations). To address this, we performed several correlation analyses between burst amplitude and burst apparent spatial width, between and across sessions. Further, we investigated whether the distribution of the propagation direction is biased by a potential bias in the spatial domain, such as the selective cancellation of bursts generated by tangential sources (*Ahlfors et al., 2010*; *Eulitz et al., 1997*). We aimed to examine the distribution of propagation directions from beta bursts which form a uniform distribution across space. To retain a nigh SNR, we focussed this analysis to the area of the ROI that has a burst probity higher than 0.44 (see *Figure 1d*). We then iteratively removed bursts until the burst distribution across this space was uniform. We found that the distribution of propagation directions using a subset of bursts forming a uniform spatial distribution is comparable to the original set of bursts. This analysis suggests that potential spatial sampling bias does not shape the propagation analysis.

Regarding the interpretation of travelling waves, there is inherent ambiguity concerning the mechanisms that generate a travelling wave (see 'Distinct anatomical propagation axes of sensorimotor beta activity'; *Prechtl et al., 2000*; *Ermentrout and Kleinfeld, 2001*). While this concerns travelling wave analyses across a range of spatial scales and recording techniques, the source space analysis employed here entails an additional issue – namely whether the propagation directions can be trivially explained by spatial variability in the LCMV weights. To address this issue, we correlated the latency

of the critical points across space before and after regressing out the main components of the spatial variability in the LCMV weights. Specifically, we performed PCA on the LCMV weights and retained the components that explained 90% of the variance in the LCMV weights. We then performed, for each critical point of each burst, a multiple regression analysis with the latencies of the critical point across space as dependent variable and the coefficients of the PCs across space as independent variables. We then correlated the latency of the critical points across space with the residuals of the multiple regression. Pearson's *r* was first averaged across critical points within bursts, and then across bursts.

## Statistical analysis

Statistical analysis was performed using nonparametric testing in Matlab2019b. If not stated otherwise, descriptive statistics depict mean and standard deviation of the median across subjects. Burst characteristics with unimodal distributions (e.g. burst amplitude, burst propagation speed) were compared using the Wilcoxon signed-rank test on the medians of the distribution. The test statistic is reported as a value of *T*. Burst characteristics with multimodal distributions (e.g. spatial location) were compared using a two-sample Kolmogorov–Smirnov test on the single subject level. The test statistic is reported as the value of *KS* (i.e. mean and range across subjects). Two circular distributions (e.g. propagation direction pre- and post-movement) were compared using two-sample Watson's $U^2$ test (*Landler et al., 2021*) on the single subject level. The test statistic is reported as the value of $U^2$ test (i.e. mean and range across subjects).

To test whether there is significant spatiotemporal structure in burst activity, we compared the propagation direction of real and surrogate data. Specifically, for a subset of bursts, that is, 100 randomly selected bursts sampled across all subjects, 1000 surrogates were created for each burst from the data after sign ambiguity was resolved (see *Figure 6*). Surrogate data were obtained by computing the discrete Fourier transform of the data, randomising the phase spectrum while preserving the amplitude spectrum, and then computing the inverse discrete Fourier transform to obtain the surrogated data (method 3 in *Hurtado et al., 2004*). For each burst, the magnitude of the propagation direction of the real data was compared to the distribution from 1000 surrogates.

To quantify the overall propagation direction, a mixture of four von Mises functions was fitted to the average of the subjects' probability distribution of propagation directions across bursts. This provides an estimate of the angle and length of the von Mises functions. Reliability of von Mises functions was assessed using a split-half reliability. In total, 500 split halves were computed and 4 von Mises functions estimated on each half independently. For both, angle and length, the difference between the two halves was computed. Further, to test whether the von Mises functions were significantly different from zero, non-parametric permutation testing was employed on the length of the von Mises functions. Permutations were carried out by randomising the propagation direction of each burst. In total, 5000 permutations were computed before statistical significance was determined on the length of the von Mises functions while correcting for multiple comparison at p<0.01.

## Software

All analyses are performed using freely available tools in MATLAB (The MathWorks Inc (2022a) version 9.12.0, Natick, Massachusetts, RRID:SCR_001622). The code carrying out the analysis in this article can be found here: https://github.com/cathazi/Zich_2023_3DBursts (copy archived at *Zich et al., 2023*). This analysis depends on a number of other toolboxes and software packages. The MEG processing was performed using the OHBA Software Library (*OHBA Analysis Group, 2017*). This builds upon Fieldtrip, SPM and FSL to provide a range of useful tools for M/EEG analyses. Further, the following MATLAB toolboxes were used: Computer Vision Toolbox (version 10.2), Image Processing Toolbox (version 11.5), Statistics and Machine Learning Toolbox (version 12.3), and Signal Processing Toolbox (version 9.0). Moreover, the MarsBaR region of interest Toolbox for SPM (*Brett et al., 2002*, version 0.45) and CircStat: A Matlab Toolbox for Circular Statistics (*Berens, 2009*; *Berens, 2012*) were used. Exact geodesic for triangular meshes (*Kirsanov, 2008*), Inhull (*D'Errico, 2012*), Point biserial correlation (*Nagel, 2006*), and Triangle/Ray Intersection (*Tuszynski, 2018*) were used. The 3D burst analyses can be very computationally intensive even on a modern computer system. The analyses in this article were computed on a MacBook Pro with a 2.6GHz 6-Core Intel Core i7 and 32Gb of RAM.

Details of the installation and setup of the dependencies can be found in the README.md file in the main study repository.

## Acknowledgements

CZ was supported by the Brain Research UK (201718-13, 201617-03). AJQ was supported by the NIHR Oxford Health Biomedical Research Centre and a Wellcome Trust Strategic Award (098369/Z/12/Z). GCO was supported by EPSRC (EP/T001046/1) funding from the Quantum Technology hub in sensing and timing (sub-award QTPRF02). LCM was supported by the Medical Research Council (MR/N013867/1). The Wellcome Centre for Human Neuroimaging and The Wellcome Centre for Integrative Neuroimaging are supported by core funding from the Wellcome Trust (203147/Z/16/Z and 203139/Z/16/Z, respectively).

## Additional information

### Funding

| Funder | Grant reference number | Author |
| --- | --- | --- |
| Brain Research UK | 201718-13 | Catharina Zich |
| Brain Research UK | 201617-03 | Catharina Zich |
| Wellcome Trust | 098369/Z/12/Z | Andrew J Quinn |
| Engineering and Physical Sciences Research Council | EP/T001046/1 | George O'Neill |
| Medical Research Council | MR/N013867/1 | Lydia C Mardell |
| Wellcome Trust | 203147/Z/16/Z and 203139/Z/16/Z | Lydia C Mardell |

The funders had no role in study design, data collection and interpretation, or the decision to submit the work for publication. For the purpose of Open Access, the authors have applied a CC BY public copyright license to any Author Accepted Manuscript version arising from this submission.

### Author contributions

Catharina Zich, Conceptualization, Formal analysis, Visualization, Methodology, Writing – original draft, Project administration; Andrew J Quinn, Conceptualization, Methodology, Writing – review and editing; James J Bonaiuto, Data curation, Investigation, Writing – review and editing; George O'Neill, Lydia C Mardell, Methodology, Writing – review and editing; Nick S Ward, Resources, Funding acquisition, Writing – review and editing; Sven Bestmann, Conceptualization, Resources, Funding acquisition, Writing – review and editing

### Author ORCIDs

Catharina Zich http://orcid.org/0000-0002-0705-9297
Andrew J Quinn http://orcid.org/0000-0003-2267-9897
Lydia C Mardell http://orcid.org/0000-0003-3180-3239
Sven Bestmann http://orcid.org/0000-0002-6867-9545

### Ethics

Human subjects: The study protocol was in full accordance with the Declaration of Helsinki, and all participants gave written informed consent after being fully informed about the purpose of the study. The study protocol, participant information, and form of consent, were approved by the UCL Research Ethics Committee (reference number 5833/001).

### Decision letter and Author response

Decision letter https://doi.org/10.7554/eLife.80160.sa1
Author response https://doi.org/10.7554/eLife.80160.sa2

## Additional files

### Supplementary files
• MDAR checklist

### Data availability
Data are available via the Open Science Framework (OSF).

The following dataset was generated:

| Author(s) | Year | Dataset title | Dataset URL | Database and Identifier |
|---|---|---|---|---|
| Zich C | 2023 | Spatiotemporal organization of human sensorimotor beta burst activity | https://doi.org/10.17605/OSF.IO/9CQHB | Open Science Framework, 10.17605/OSF.IO/9CQHB |

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
