## [Editor Report]

This paper provides important insights into the spatial organization of β-oscillatory activity in the human brain, which is a crucial dynamic feature of frontal and parietal networks involved in movement preparation and sensory prediction. Using high-resolution source reconstruction with Magnetoencephalography in humans, the authors provide compelling evidence demonstrating that β oscillations are organized as travelling waves in two distinct directions relative to the central sulcus. Furthermore, the study convincingly shows that the spatiotemporal organization of β bursts is systematically linked to behavior, specifically motor execution. These findings have important implications for our understanding of the neural mechanisms that underlie movement planning and execution in the human brain.

---

## [Decision Letter]

**Decision letter after peer review:**

Thank you for submitting your article "Spatiotemporal organization of human sensorimotor β burst activity" for consideration by *eLife*. Your article has been reviewed by 3 peer reviewers, and the evaluation has been overseen by a Reviewing Editor and Chris Baker as the Senior Editor. The following individual involved in review of your submission has agreed to reveal their identity: Robert Law (Reviewer #3).

The reviewers have discussed their reviews with one another, and the Reviewing Editor has drafted this to help you prepare a revised submission. As you can see from the reviews, there are a number of methodological/technical concerns that need to be addressed, although the reviewers considered the work generally sound in terms of methods. A more critical point is the nature of the conceptual advance in terms of understanding the mechanisms or functions of β bursts and waves. We expect the authors to expand their study in terms of the significance of the findings, by performing new analyses and analyzing new data. Thus, we are happy to invite a revision, but we expect new data and/or analyses that substantially improve the significance of the manuscript.

Essential revisions:

As you can see from the reviewers, the study received mixed evaluations. We focus here on comments pertaining to the technical aspects (comments 1-9), and the conceptual advance (comment 10).

1) Could the beamforming weights transform any activity (regardless of its original structure) into a profile that shows travelling waves? The authors do report a high correlation. Please clarify.

2) It seems that the authors do the analysis in the embedding space created by the 2D visualizer. To what extent does this bias the results? Why was the analysis not done in the 3D space?

3) Could there be a trivial explanation for the two wave directions that are reported. MEG is of course sensitive mostly to tangential current flow. This tangential current flow comes from the walls of the sulci or cortical tissue that folds around the midline or laterally. If you look at the orientation of the walls of the sulci then this predicts current flow in the anterior to posterior plane. And the medial/lateral axis as well.

4) Modifying the threshold at which β events are detected could alter their reported properties and expression in space and time. The authors should therefore perform parameter sweeps on e.g. the thresholds for detection of oscillation bursts to determine whether their conclusions on β properties and propagation hold. If this additional analysis does not change their story, it would lend confidence to the results/conclusions.

5) Determining the generators of β events at different locations is a tricky issue. The authors mentioned a single generator that is responsible for propagating β along the two axes described. However, it is not clear through what mechanism the β events could travel along the neural substrate without additional local generators along the way. Previous work on β events examined how a sequence of synaptic inputs to supra and

infragranular layers would contribute to a typical β event waveform. Although it is possible other mechanisms exist, how might this work as the β events propagate through space? Some further explanation/investigation on these issues is therefore warranted.

6) Given that the main angle of this paper is methodological, it would be important to provide the source code on github along with documentation and a standard "notebook".

7) The authors should describe in more detail the properties of the β bursts in both directions: Are there any distinguishing features for the β bursts that propagate along each primary direction? How far do each of the traveling waves go in each direction? Is the

frequency and properties nearly constant as the waves travel or do the dominant frequencies show any shifts the farther the waves get from their generator? If there is a single generator responsible for the β waves that travel in each direction, then through

what mechanism are the β events created as they propagate further from their source?

8) Introduction:

"Finally, our results suggest that sensorimotor β bursts occurring before and after a movement share the same generator but can be distinguished by their anatomical, spectral and spatiotemporal characteristics, indicating distinct functional roles." If β bursts before/after movements have the same generator, what distinguishes the initiation of movement vs lack of movement? As a result of their work here, would the authors hypothesize that β contributes to either generation or suppression of movement?

9) Spatial cancellation of β occurring simultaneously on both sides of the central sulcus may occur (these events would be near to one another and with opposite orientations, cancelling out the magnetic field, see Ahlfors et al. 2010). This would ultimately lead to fewer detected events on the sulcal walls, compared with the sulcal banks. Ultimately the study may be throwing out a lot of ground-truth events from the walls, and the fact that the authors keep more events from deep in the sulcus will bias the traveling wave analysis that follows.

In Figure 1, the authors seem to show a higher density of β events relatively deep in the sulcus compared to the sulcal walls. This is certainly an interesting result if true! But even given only the occasional synchronization of mesoscale cortical neighborhoods, it appears that events in the sulcal walls will still be systematically undersampled and those deep in the sulcus oversampled here, by vice or virtue of cortical geometry as it pertains to the magnetic field.

This spatial sampling bias could impact nearly all aspects of the event propagation analysis that follows, and so must be considered in sufficient detail.

A "worst-case scenario" is that ground-truth events may be uniformly distributed across much of the region of interest, and where both the reported directionality and relative incidence rates of traveling wave classes are artefactually biased. Treating this potential issue could, for instance, come in the form of (1) developing further controls and/or (2) repeating the analysis after resampling events to reflect a potential worst-case ground-truth scenario.

10) While the reviewers acknowledged the methodological advance, they were not sufficiently convinced that the present study provides sufficient functional advance. For example, no meaningful difference was found between bursts traveling along the two different principal modes of propagation, and importantly, no relation with behavior (response time) was found. The same stands for pre vs. post motor bursts, except for the expected finding that post-motor bursts are more frequent and tend to be of greater amplitude (yielding the observation of a so-called β rebound, on average across trials).

---

## [Author Response]

Essential revisions:As you can see from the reviewers, the study received mixed evaluations. We focus here on comments pertaining to the technical aspects (comments 1-9), and the conceptual advance (comment 10).1) Could the beamforming weights transform any activity (regardless of its original structure) into a profile that shows travelling waves? The authors do report a high correlation. Please clarify.

This is an important issue, which we investigated in different ways.

First, as outlined in the paper: To examine whether the two main propagation axes can be trivially explained by spatial variability in the beamformer weights, we correlated the latency of the critical points across space before and after regressing out the main components of the spatial variability in the LCMV weights. Specifically, we performed PCA on the LCMV weights and retained the components that explained 90% of the variance in the LCMV weights. We then performed, for each critical point of each burst, a multiple regression analysis with the latencies of the critical point across space as the dependent variable, and the coefficients of the PCs across space as independent variables. We then correlated the latency of the critical points across space with the residuals of the multiple regression. Pearson’s r was first averaged across critical points within bursts, and then across bursts. We found significant correlations (Pearson’s r: M = 0.61, SD = 0.27 across individuals, all p’s < 0.05). This result suggests that beamformer weights contribute to, but do not solely explain the observed propagation directions.

Second, the LCMV weights are fixed within a recording session, yet we still observe considerable variability in the propagation direction both within and across bursts. If the directions would be predominantly driven by the LCMV weights, we would expect low moment-to-moment variability in the propagation direction.

Third, we now conducted the same propagation analysis but for a different frequency range (γ, γ, 60-90Hz) for one exemplar subject. Please see Author response image 1. As the filter was applied before the beamformer, we have a separate set of LCMV weights for the β and γ frequency range. We inspected the structure of the PCA components that explained 95% of the variance in the LCMV weights and leadfields for the β and γ frequency range. We found that the structure of these PCs was comparable cross the two frequency ranges (Author response image 1). Therefore, we reason that: if the beamformer solely explains the propagation directions, the distribution of propagation directions should be comparable between the β and γ frequency range, given the comparable structure of the PCs. However, we find that, in contrast to the β frequency range, for the γ range no dominant directions were observed and von Mises functions did not converge. This suggests that the LCMV weights do not lead to an unavoidable phase-gradient and appearance of propagation.

Together we believe this set of analysis indicated that beamformer weights contribute to, but do not solely explain the observed propagation directions.

**Author response image 1. sa2fig1:** Propagation direction and LCMV weights for the β and γ frequency range. (**a**) Propagation direction for the β (β, 13-30Hz) and γ (γ, 60-90) frequency range for one exemplar subject (across all sessions of that subject). (**b**) First three PCs of the LCMV beamformer leadfields for the β (left) and γ (right) frequency range for one session from the same subject as in a visualised on the folded and inflated.

2) It seems that the authors do the analysis in the embedding space created by the 2D visualizer. To what extent does this bias the results? Why was the analysis not done in the 3D space?

We would like to clarify that most of the analyses were performed in 3D space on the original (i.e., folded) surface. The only exception is the quantification of the propagation direction and propagation speed.

We estimated the propagation direction on the inflated surface for easier interpretability and consistency with published reports using multi-electrode arrays (Balasubramanian et al., 2020; Best et al., 2016; Rubino et al., 2006; Takahashi et al., 2011, 2015). One of the challenges of estimating propagation direction in 3D on the original folded surface is illustrated in Author response image 2. Here, two bursts, one on the crown (a) and one on the bank (b) of the precentral gyrus, are projected onto the original (bottom) and the inflated (top) surface. For the burst on the crown the propagation direction (illustrated by the colour gradient, and black arrow) is primarily along the x- and y-axis for both the original and inflated surface. However, for the burst on the bank of the gyrus, the propagation direction is primarily along the z-axis on the original surface, and along the y-axis of the inflated surface.

Thus, if the propagation direction is estimated on the original surface, the spatial location and folding need to be taken into consideration. This is especially important here, where we analyse bursts at a relatively small spatial scale, with bursts often spanning a relatively small area of the cortex. In contrast, if propagation is estimated across larger areas spanning across several sulci (as is the case, for example, in many EEG studies) the x- and y-axes will naturally dominate the propagation direction, and this point becomes less critical.

The spatial folding could be considered by estimating the propagation direction from the original surface as a function of the surface normals. However, we opted for a more intuitive approach that enabled us to easily relate the results from our non-invasive recordings with results from invasive multi-electrode arrays, and provide more intuitive ways for visualization.

We also note that our results were not simply just projected onto a 2D visualizer. Rather, the inflated cortical surface preserves the spatial relationship among vertices and allows for a direct mapping between the 3D and 2D data projections through a corresponding transformation matrix.

Regarding propagation speed, we estimated speed using both the distance derived from the original cortical surfaces and the inflated surfaces. Despite absolute differences between the two estimates (i.e., propagation speed derived from the original surfaces is roughly twice as fast as the propagation speed derived from the inflated surface, Figure 2—figure supplement 2), their relative effects (i.e., the difference in propagation speed between the two dominant axes, Figure 4b and the difference in propagation speed between pre- and post-movement bursts Figure 5b) are consistent.

**Author response image 2. sa2fig2:** Propagation direction estimated from the original (i.e., folded) and inflated surface. (**a**) Burst on the crown of the precentral gyrus visualised on the inflated (top) and original (bottom) surface. Colour denotes the relative latency of the neural activity at a critical point and the arrow indicates the estimated direction of travel (as described in the Methods section of the manuscript as well as Figure 2). (**b**) Same as (a) for a burst on the bank of the precentral gyrus.

3) Could there be a trivial explanation for the two wave directions that are reported. MEG is of course sensitive mostly to tangential current flow. This tangential current flow comes from the walls of the sulci or cortical tissue that folds around the midline or laterally. If you look at the orientation of the walls of the sulci then this predicts current flow in the anterior to posterior plane. And the medial/lateral axis as well.

The reviewers raise an interesting point here, i.e., whether the two propagation directions can be simply explained by cortical folding. While we cannot rule out a contribution of cortical folding, we would like to highlight three aspects suggesting that cortical folding does not solely explain the spatiotemporal features of bursts.

First, we have now compared the propagation directions of deeper (e.g., bursts located in a sulcus) versus shallower bursts (e.g., bursts located around the gyral crown). Cortical depth was determined using the spatial location PC 3^1^ (as shown in Figure 4—figure supplement 3 and Author response image 3). As shown in Author response image 3, the propagation direction is comparable for deeper and shallower bursts. Specifically, we found that propagation directions were not significantly different between deeper and shallower bursts (U2 [test statistic for Watson’s U2 test, see Statistical analysis]: M = 0.092 across individuals, range = 0.037 – 0.227; 8/8 p’s > 0.1). Moreover, the directions of activity in deeper bursts (a-p axis: 68/244deg; m-l axis: 141/320deg) did not differ from the directions observed in shallower bursts (a-p axis: 63/250deg; m-l axis: 143/325deg).

(^1^As a reminder, the spatial centre of bursts was described using principal components (PC). Burst distribution was characterised along the anterior/lateral - posterior/medial axis by PC 1, along the anterior/medial - posterior/lateral axis axis by PC 2, and along the z-orientation by PC 3.)

Second, and related, the cortical folding and the orientation of the cortical surface normals with regards to the sensors is highly variable both within (i.e., the angle of the central sulcus to a given sensor varies along the central sulcus) and across subjects (i.e., the architecture of the central sulcus is different across individuals, see Author response image 3). This means that the distinction between tangential = wall vs lateral = crown current flow is not clearly borne out neither within nor across subjects. We also note that more bursts were observed in the depth of the central sulcus, a point we revisit in response to reviewer 3.

Third, recordings from invasive multi-electrode arrays have shown a similar dominance of these two propagation axes. For example, Takahashi and colleagues reported that β activity in M1 of a tetraplegic patient propagated along the medial-lateral axis (Takahashi et al., 2011). In non-human primates, β activity propagates along the anterior-posterior axis in M1 (Balasubramanian et al., 2020; Best et al., 2016; Rubino et al., 2006; Takahashi et al., 2011, 2015), and along the medial-lateral axis in the dorsal premotor cortex (Rubino et al., 2006), indicating regional differences in spatiotemporal patterns (Rubino et al., 2006; Rule et al., 2018).

Together, we believe this provides convergent evidence that the two main propagation directions are not simply an artefact of cortical folding.

**Author response image 3. sa2fig3:** Cortical folding and propagation direction. (**a**) Surface plot, on the inflated surface, of cortical depth, i.e. spatial location principal component (PC) 3 (see Figure 4—figure supplement 3, for PC 1 and PC 2). CS, Central Sulcus. S1, Primary Sensory Cortex. M1, Primary Motor Cortex. PM, Premotor Cortex. (**b**) Propagation direction does not differ as a function of cortical depth. Polar probability histograms are shown for deep (brown) and shallow (cyan) bursts. Probability distributions were calculated for each subject individually and then averaged (dark line). Variance across subjects is expressed as standard deviation from the mean (shaded area). Von Mises functions were fitted separately for deep and shallow bursts. (**c**) Inter-individual differences in cortical folding in the motor area. Three exemplary surfaces are shown. For the region of interest cortical depth is colour-coded (note that on the folded surface only the crown, i.e., cyan, parts are visible).

4) Modifying the threshold at which β events are detected could alter their reported properties and expression in space and time. The authors should therefore perform parameter sweeps on e.g. the thresholds for detection of oscillation bursts to determine whether their conclusions on β properties and propagation hold. If this additional analysis does not change their story, it would lend confidence to the results/conclusions.

We thank the reviewing team for this comment. As suggested, we evaluated the effect of different burst thresholds on the burst parameters.

The threshold in the main analysis was determined empirically from the data, as in previous work (Little et al., 2019). Specifically, trial-wise power was correlated with the burst probability across a range of different threshold values (from median to median plus seven standard deviations (std), in steps of 0.25, see Figure 6—figure supplement 1). The threshold value that retained the highest correlation between trial-wise power and burst probability was used to binarize the data.

We repeated our original analysis using four additional thresholds, i.e., original threshold 0.5 std, -0.25 std, +0.25 std, +0.5 std. As one would expect, burst threshold is negatively related to the number of bursts (i.e., higher thresholds yield fewer bursts, Figure 6—figure supplement 2a [top]), and positively related to burst amplitude (i.e., higher thresholds yield higher burst amplitudes, Figure 6—figure supplement 2a [bottom]).

Similarly, the temporal duration of bursts and apparent spatial width are modulated by the burst threshold: lowering the threshold leads to longer temporal duration and larger apparent spatial width while increasing the threshold leads to shorter temporal duration and smaller apparent spatial width Figure 6—figure supplement 2b. Note that for the temporal and spectral burst characteristics, the difference to the original threshold can be numerically zero, i.e., changing the burst threshold did not lead to changes exceeding the temporal and spectral resolution of the applied time-frequency transformation (i.e., 200ms and 1Hz respectively).

Importantly, across these threshold values, the propagation direction and propagation speed remain comparable Figure 6—figure supplement 2c.

We refer to this analysis in the manuscript (page 28 line 717).

“To explore the robustness of the results analyses were repeated using a range of thresholds (Figure 6—figure supplement 2).”

5) Determining the generators of β events at different locations is a tricky issue. The authors mentioned a single generator that is responsible for propagating β along the two axes described. However, it is not clear through what mechanism the β events could travel along the neural substrate without additional local generators along the way. Previous work on β events examined how a sequence of synaptic inputs to supra andinfragranular layers would contribute to a typical β event waveform. Although it is possible other mechanisms exist, how might this work as the β events propagate through space? Some further explanation/investigation on these issues is therefore warranted.

Based on this and other comments (i.e., comments 7 and 8) we re-evaluated the use of the term ‘generator’ in this manuscript.

While the term generator can be used across scales, from micro- to macroscale, ifor the purpose of the present paper, we believe one should differentiate at least two concepts: (a) generator of β bursts, and (b) generator of travelling waves.

We realised that in the previous version of the manuscript the term ‘generator’ was at times used without context. We removed the term where no longer necessary.

Further, the previous version of the manuscript discussed putative generators of travelling waves (page 19F.) but not generators of β bursts. We now address this as follows:

“Studies using biophysical modelling have proposed that β bursts are generated by a broad infragranular excitatory synaptic drive temporally aligned with a strong supragranular synaptic drive (Law et al., 2022; Neymotin et al., 2020; Sherman et al., 2016; Shin et al., 2017) whereby layer specific inhibition acts to stabilise β bursts in the temporal domain (West et al., 2023). The supragranular drive is thought to originate in the thalamus (E. G. Jones, 1998, 2001; Mo and Sherman, 2019; Seedat et al., 2020), indicating thalamocortical mechanisms (page 22f).”

6) Given that the main angle of this paper is methodological, it would be important to provide the source code on github along with documentation and a standard "notebook".

All analyses are performed using freely available tools in MATLAB. The code carrying out the analysis in this paper can be found here: [link provided upon acceptance]. The 3D burst analyses can be very computationally intensive even on a modern computer system. The analyses in this paper were computed on a MacBook Pro with a 2.6 GHz 6-Core Intel Core i7 and 32 Gb of RAM. Details on the installation and setup of the dependencies can be found in the README.md file in the main study repository.

This information has been added to the paper in the methods section on page 35.

7) The authors should describe in more detail the properties of the β bursts in both directions: Are there any distinguishing features for the β bursts that propagate along each primary direction? How far do each of the traveling waves go in each direction? Is thefrequency and properties nearly constant as the waves travel or do the dominant frequencies show any shifts the farther the waves get from their generator? If there is a single generator responsible for the β waves that travel in each direction, then throughwhat mechanism are the β events created as they propagate further from their source?

Based on our previous work (Zich et al., 2020), we consider first and second level burst characteristics. First level burst characteristics are obtained for each burst and each domain separately (see Figure 1). For the temporal domain, the temporal on- and offset, temporal duration and temporal centre (i.e., mean of on- and offset) of each burst were obtained. Equally, low and high frequency boundaries, frequency spread and frequency centre (i.e., mean of low and high boundary) were extracted for the spectral domain from each burst. For the spatial domain, we obtained the spatial width (i.e., the total surface area defined as the sum of the area of all faces), the size in each dimension (x, y, z) using the minimum bounding rectangle (i.e., bounding box), and the spatial centre.

These first level burst characteristics form the basis for second level burst characteristics, which can be broadly summarised as (a) combinations and (b) interactions of characteristics within and across domains. For example, we combined temporal duration, frequency spread, and apparent spatial width into a single metric, i.e., burst extent. Further we explored the spatiotemporal interaction in depth.

One of the critical questions of this paper is whether bursts activity that propagate along the a-p axis or m-l axis differ systematically in any of these first or second level burst characteristics (see page 13f, Figure 4). This is relevant because distinct properties of bursts in different locations or travel directions may be of functional relevance or may be underpinned by distinct mechanisms.

In addition to this analysis of burst characteristics, the reviewers highlighted further signal properties that could be explored, e.g., variability of signal properties within a burst. We think that these signal properties are beyond the scope of the present paper in that the variability and temporal stability of bursts is an interesting question that warrant detailed exploration. For this reason, we now mention this in the manuscript (page 30), as follows:

“These burst metrics are not exhaustive. Other 2nd level burst characteristics include the interaction between frequency centre and spatial location (Zich et al., 2020), waveform variability (Szul et al., 2022), and the variability of the 1st level burst characteristics, such as the variability in (instantaneous) frequency, within and across bursts. Future work may address the variability and temporal stability of bursts, for example using methods such as empirical mode decomposition (Huang et al., 1998).”

Regarding the aspects concerning the ‘generator’ please refer to our answer to point 7.

8) Introduction:"Finally, our results suggest that sensorimotor β bursts occurring before and after a movement share the same generator but can be distinguished by their anatomical, spectral and spatiotemporal characteristics, indicating distinct functional roles." If β bursts before/after movements have the same generator, what distinguishes the initiation of movement vs lack of movement? As a result of their work here, would the authors hypothesize that β contributes to either generation or suppression of movement?

Based on this and other comments (i.e., comments 5 and 7) we re-evaluated the use of the term ‘generator’ in this manuscript. Please see comment 5 for more details. The paragraph in the introduction now reads:

“Finally, our results suggest that sensorimotor β bursts occurring before and after a movement can be distinguished by their anatomical, spectral and spatiotemporal characteristics, indicating distinct functional roles.”

9) Spatial cancellation of β occurring simultaneously on both sides of the central sulcus may occur (these events would be near to one another and with opposite orientations, cancelling out the magnetic field, see Ahlfors et al. 2010). This would ultimately lead to fewer detected events on the sulcal walls, compared with the sulcal banks. Ultimately the study may be throwing out a lot of ground-truth events from the walls, and the fact that the authors keep more events from deep in the sulcus will bias the traveling wave analysis that follows.In Figure 1, the authors seem to show a higher density of β events relatively deep in the sulcus compared to the sulcal walls. This is certainly an interesting result if true! But even given only the occasional synchronization of mesoscale cortical neighborhoods, it appears that events in the sulcal walls will still be systematically undersampled and those deep in the sulcus oversampled here, by vice or virtue of cortical geometry as it pertains to the magnetic field.This spatial sampling bias could impact nearly all aspects of the event propagation analysis that follows, and so must be considered in sufficient detail.A "worst-case scenario" is that ground-truth events may be uniformly distributed across much of the region of interest, and where both the reported directionality and relative incidence rates of traveling wave classes are artefactually biased. Treating this potential issue could, for instance, come in the form of (1) developing further controls and/or (2) repeating the analysis after resampling events to reflect a potential worst-case ground-truth scenario.

Thank you for highlighting this point. As suggested, we performed a set of control analysis to investigate this issue.

First, in line with our response to comment #3 we compared the propagation directions of deeper (e.g., bursts located in a sulcus) versus shallower bursts (e.g., bursts located around the gyral crown). Cortical depth was determined using the spatial location PC 3^2^ (visualised in Figure 4—figure supplement 3 and Author response image 3). As shown in Author response image 3, the distribution of propagation direction for deeper and shallower bursts do not significantly differ between deeper and shallower bursts (U2 [test statistic for Watson’s U2 test, see Statistical analysis]: M = 0.092 across individuals, range = 0.037 – 0.227; 8/8 p’s > 0.1). Moreover, the directions of activity in deeper bursts (a-p axis: 68/244deg; m-l axis: 141/320deg) did not differ from the directions observed in shallower bursts (a-p axis: 63/250deg; m-l axis: 143/325deg). This analysis indicates that the propagation direction is not modulated by cortical depth.

Further, we explicitly investigate the outlined ‘worst-case scenario’. To this end, we repeated our main analysis considering a subset of bursts that yields a uniform distribution across much of the ROI, as suggested by the reviewer. In order to use as many bursts as possible for this analysis, we limited the analysis to the area of the ROI that has a burst probability higher than 0.44 (Author response image 4). We then iteratively removed bursts until a uniform distribution of bursts across space was achieved (Author response image 4). Specifically, for each iteration we determined the local maxima of the burst probability map, and removed the burst which spatial centre was closest to the local maxima. Thus, the remaining subset of bursts results in a uniform distribution of burst probability across space. Using only these bursts we generated the polar probability histogram as shown in Figure 4a. We found that the distribution of propagation directions is comparable to the original set of bursts (Author response image 4). This analysis suggests that potential spatial sampling bias does not bias the propagation analysis.

Together we believe that even if spatial cancellation occurred the propagation analysis is robust to this. We now include the reasoning and the second control analysis in the manuscript (page 13f and 33f).

(^2^As a reminder, the spatial centre of bursts was described using principal components (PC). Burst distribution was characterised along the anterior/lateral - posterior/medial axis by PC 1, along the anterior/medial - posterior/lateral axis by PC 2, and along the z-orientation by PC 3.)

**Author response image 4. sa2fig4:** Propagation direction of a subsample of bursts. (**a**) Burst probability as a function of space across all bursts of all subjects on the inflated surface (top) and original surface (bottom). Unlike Figure 2d we focus here on the area of the ROI that has a burst probability higher than 0.44. Highlighted are the central sulcus and the borders for 0.44 and 0.49 burst probability. (**b**) Burst probability of a subset of burst forming a uniform spatial distribution across the area of the ROI that has a burst probability higher than 0.44. (**c**) Polar probability histogram for the subset of bursts showing the probability distribution of burst direction in MNI space. Probability distributions were calculated for each subject individually and then averaged (dark grey line). Variability across subjects is expressed as standard deviation from the mean (light grey area). To estimate the dominant propagation directions, a mixture of von Mises functions was fitted to the averaged probability distribution (arrows). The four functions lie on two axes. One axis has an anterior-posterior orientation which is approximately perpendicular to the orientation of the central sulcus (a-p), while the other axis runs in approximately medial-lateral orientation which is approximately parallel to the orientation of the central sulcus (m-l). Similar to Figure 4a.

10) While the reviewers acknowledged the methodological advance, they were not sufficiently convinced that the present study provides sufficient functional advance. For example, no meaningful difference was found between bursts traveling along the two different principal modes of propagation, and importantly, no relation with behavior (response time) was found. The same stands for pre vs. post motor bursts, except for the expected finding that post-motor bursts are more frequent and tend to be of greater amplitude (yielding the observation of a so-called β rebound, on average across trials).

In response to this comment, we would like to highlight two important points.

First, our work constitutes the first non-invasive human confirmation of invasive work in animals (Balasubramanian et al., 2020; Roberts et al., 2019; Rule et al., 2018; Balasubramanian et al., 2020; Best et al., 2016; Rubino et al., 2006; Takahashi et al., 2011, 2015) and patients (Takahashi et al., 2011). Thus, these results bridges between recordings limited to the size of multielectrode arrays (roughly 0.16 cm^2^; Balasubramanian et al., 2020; Best et al., 2016; Rubino et al., 2006; Takahashi et al., 2011, 2015) and human EEG recordings spanning across large areas of the cortex and several functionally distinct regions (Alexander et al., 2016; Stolk et al., 2019). The ability to access these neural signatures noninvasively is important for cross-species comparison. This further enables us, to provide an in-depth analysis of the spatiotemporal diversity of human MEG signals and a detailed characterisation of the two propagation directions, which significantly extends previous reports. We note that their functional role remains undetermined also in these animal studies, but being able to identify these signals now in humans can provide a steppingstone for identifying their role.

Second, and related, the reviewers are correct that we did not observe distinct propagation directions between pre- and post-movement bursts, nor a relationship with reaction time. However, such a null result would be relevant, in our view, towards understanding what the functional relevance of these signals, if any, might be. Recent work in macaques indicates that the spatiotemporal patterns of high-γ activity carry kinematic information about the upcoming movement (Liang et al. 2023). The functional role of β may therefore be more complex and not relate to reaction times or kinematics in a straightforward manner. We believe this is a relevant observation, and in keeping with the continued efforts to identify how sensorimotor β relates to behaviour. It is increasingly clear that spatiotemporal diversity in animal recordings and human E/MEG and intracranial recordings can constitute a substantial proportion of the measured dynamics. As such, our report is relevant in narrowing down what these signals may reflect.

Together, we think that our work provides new insights into the multidimensional and propagating features of burst activity. This is important for the entire electrophysiology community, as it transforms how we commonly analyse and interpret these important brain signals. We anticipate that our work will guide and inspire future work on the mechanistic underpinnings of these dominant neural signals. We are confident that our article has the scope to reach out to the diverse readership of *eLife*.